# Lactate transporter MCT1 in hepatic stellate cells promotes fibrotic collagen expression in nonalcoholic steatohepatitis

Kyounghee Min[1], Batuhan Yenilmez[1], Mark Kelly[1], Dimas Echeverria[2], Michael Elleby[1], Lawrence M Lifshitz[1], Naideline Raymond[1], Emmanouela Tsagkaraki[1], Shauna M Harney[1], Chloe DiMarzio[1], Hui Wang[1], Nicholas McHugh[2], Brianna Bramato[2], Brett Morrison[3], Jeffery D Rothstein[3], Anastasia Khvorova[2], Michael P Czech[1]*

[1]Program in Molecular Medicine, University of Massachusetts Chan Medical School, Worcester, United States; [2]RNA Therapeutics Institute, University of Massachusetts Chan Medical School, Worcester, United States; [3]Department of Neurology, Johns Hopkins School of Medicine, Baltimore, United States

*For correspondence:
michael.czech@umassmed.edu

**Abstract** Circulating lactate is a fuel source for liver metabolism but may exacerbate metabolic diseases such as nonalcoholic steatohepatitis (NASH). Indeed, haploinsufficiency of lactate transporter monocarboxylate transporter 1 (MCT1) in mice reportedly promotes resistance to hepatic steatosis and inflammation. Here, we used adeno-associated virus (AAV) vectors to deliver thyroxin binding globulin (TBG)-Cre or lecithin-retinol acyltransferase (Lrat)-Cre to MCT1$^{fl/fl}$ mice on a choline-deficient, high-fat NASH diet to deplete hepatocyte or stellate cell MCT1, respectively. Stellate cell MCT1KO (AAV-Lrat-Cre) attenuated liver type 1 collagen protein expression and caused a downward trend in trichrome staining. MCT1 depletion in cultured human LX2 stellate cells also diminished collagen 1 protein expression. Tetra-ethylenglycol-cholesterol (Chol)-conjugated siRNAs, which enter all hepatic cell types, and hepatocyte-selective tri-N-acetyl galactosamine (GN)-conjugated siRNAs were then used to evaluate MCT1 function in a genetically obese NASH mouse model. MCT1 silencing by Chol-siRNA decreased liver collagen 1 levels, while hepatocyte-selective MCT1 depletion by AAV-TBG-Cre or by GN-siRNA unexpectedly increased collagen 1 and total fibrosis without effect on triglyceride accumulation. These findings demonstrate that stellate cell lactate transporter MCT1 significantly contributes to liver fibrosis through increased collagen 1 protein expression in vitro and in vivo, while hepatocyte MCT1 appears not to be an attractive therapeutic target for NASH.

## eLife assessment

This **convincing** manuscript represents a **valuable** advance in understanding the role of MCT1 – a transporter for lactate and other organic anions – in hepatocytes and hepatic stellate cells in the liver. The authors also generate exciting new tools to investigate hepatic stellate cell biology, and these may have much broader applications, but future studies are required to validate these new tools.

## Introduction

Nonalcoholic fatty liver disease (NAFLD) is the most common chronic liver disease, afflicting over a quarter of the world's population. It describes a spectrum of liver diseases ranging from simple

steatosis to nonalcoholic steatohepatitis (NASH) (*Younossi et al., 2016*; *Estes et al., 2018*). Steatosis is considered relatively benign as lifestyle modifications can reverse fatty liver to a healthy condition. On the other hand, NASH is characterized by severe steatosis, inflammation, and fibrosis. Severe fibrotic stages of NASH can develop into permanent liver damage, and the disease can progress to cirrhosis and hepatoma. Currently, NASH is a leading cause for liver transplantation, and there is no FDA-approved therapeutic for NASH (*Estes et al., 2018*; *Alexander et al., 2019*; *Diehl and Day, 2017*; *Friedman et al., 2018*). Since NASH is closely associated with the hallmarks of type 2 diabetes and obesity, including chronic overnutrition, insulin resistance, and dyslipidemia, it is likely that substrate overload to the liver contributes to its cause (*Friedman et al., 2018*). Accordingly, NAFLD and NASH patients display hyperactive liver tricarboxylic acid (TCA) cycle flux due to an over-abundance of upstream metabolites (*García-Ruiz and Fernández-Checa, 2018*; *Sunny et al., 2017*). These findings suggest that lowering substrate influx to the liver is a promising strategy to prevent and possibly alleviate steatosis and NASH.

Human subjects with type 2 diabetes and obesity reportedly have increased plasma lactate levels (*Lovejoy et al., 1992*; *Crawford et al., 2010*; *Juraschek et al., 2013a*; *Juraschek et al., 2013b*; *DiGirolamo et al., 1992*; *Sabater et al., 2014*; *Jansson et al., 1994*). Lactate is produced and released into the circulation when cellular glycolytic flux surpasses mitochondrial oxidative capacity. Once considered to be simply a metabolic waste product, lactate is now recognized as a primary fuel for the TCA cycle in liver and thus an essential energy source (*Hui et al., 2017*; *Rabinowitz and Enerbäck, 2020*). Additionally, it is a critical regulator that contributes to whole-body energy homeostasis (*Brooks, 2020*; *Li et al., 2022*). Under physiological conditions, cellular lactate levels are tightly controlled by monocarboxylate transporters (MCTs). MCTs are members of the solute carrier 16A (SLC16A) family, which are proton-coupled transmembrane protein transporters. Among 14 MCT isoforms, only MCTs 1–4 have been shown to transport monocarboxylate molecules such as lactate, pyruvate, short-chain fatty acids, and ketone bodies (*Felmlee et al., 2020*). Notably, MCT1 is denoted as a primary lactate transporter as it is the most widely distributed MCT isoform in various metabolic tissues and has a high affinity for lactate, maintaining basal cellular homeostasis according to transmembrane lactate gradients (*Li et al., 2022*; *Halestrap, 2013*). Reportedly, MCT1 haploinsufficiency in mice reduces MCT1 protein levels to nearly half in major metabolic tissues such as liver, brain, and white adipose tissues, and these mice are resistant to diet-induced obesity and liver steatosis and inflammation (*Lengacher et al., 2013*; *Carneiro et al., 2017*; *Hadjihambi et al., 2023*). The role of MCT1 in hypothalamus and adipose tissues in these phenotypes was ruled out, as selective MCT1 depletion in those tissues either increased food intake and body weight (*Elizondo-Vega et al., 2016*) or enhanced systemic inflammation and insulin resistance (*Lin et al., 2022*). Thus, the question of which tissue or tissues are responsible for the phenotype of whole-body MCT1 haploinsufficiency is not solved.

The above considerations suggest the possibility that MCT1KO in one or more liver cell types may explain the effects of MCT1 haploinsufficiency in mice. Since hepatocytes account for the majority of liver cells and have high rates of lipogenesis and triglyceride (TG) accumulation, lactate levels governed by hepatocyte MCT1 could be involved in regulating steatosis. On the other hand, while hepatic stellate cells account for only 5–10% of the hepatic cell population, they are the major cell type contributing to hepatic fibrogenesis (*Wake, 1971*; *Mederacke et al., 2013*). Fate tracing studies have revealed that 82–96% of myofibroblasts are derived from hepatic stellate cells, which are liver-specific pericytes (*Mederacke et al., 2013*). During NASH progression, multiple liver injury signals stimulate the transition of vitamin A-storing quiescent hepatic stellate cells into fibrogenic, proliferative myofibroblasts that produce and secrete collagen fibers (*Trautwein et al., 2015*; *Lee et al., 2015*; *Puche et al., 2013*; *Tsuchida and Friedman, 2017*). As a result, healthy hepatic parenchyma is replaced with a collagen-rich extracellular matrix, turning the liver into a hardened and scarred tissue (*Mehal et al., 2011*). In general, major organ fibrosis is directly correlated with morbidity and mortality, contributing up to 45% of deaths in developed countries (*Wynn, 2008*). Thus, targeting activated hepatic stellate cells has become a major strategy in NASH therapeutics development (*Friedman et al., 2018*; *Tsuchida and Friedman, 2017*). However, the role of MCT1 in hepatic stellate cells activation or fibrogenesis has not been investigated.

The aim of the present studies was to investigate the role of hepatic lactate transport via MCT1 in lipid metabolism and fibrogenesis in NASH, and to determine its potential suitability as a therapeutic target. Two key unanswered questions were of particular interest: (1) is it hepatocyte-specific MCT1

depletion that protects mice with MCT1 haploinsufficiency from liver lactate overload and NAFLD and (2) does liver stellate cell MCT1 promote hepatic fibrogenesis that occurs in NASH? We tested the possible enhancement of lipogenesis and fat accumulation via MCT1 function specifically in hepatocytes using adeno-associated virus (AAV)-mediated thyroxin binding globulin (TBG)-Cre MCT1KO in MCT1$^{fl/fl}$ mice, and in other experiments by silencing hepatocyte MCT1 with tri-*N*-acetyl galactosamine (GN)-conjugated siRNA. These experiments showed that hepatocyte MCT1 loss decreased expression of enzymes in the de novo lipogenesis (DNL) pathway, but did not diminish overall steatosis. Surprisingly, hepatocyte MCT1KO increased liver fibrosis in two mouse models of NASH. In contrast, hepatic stellate cell-selective MCT1KO, achieved by injection of AAV9-lecithin-retinol acyltransferase (Lrat)-Cre into MCT1$^{fl/fl}$ mice, did attenuate collagen production and fibrosis. Our findings underscore the critical importance of implementing cell type-specific targeting strategies to diminish NASH fibrogenesis.

## Results

### MCT1 depletion prevents TGF-β1-stimulated type 1 collagen production in cultured human LX2 stellate cells

As fate tracing studies have revealed that 82–96% of myofibroblasts are derived from hepatic stellate cells (*Mederacke et al., 2013*), we employed a simple in vitro system utilizing LX2 human hepatic stellate cells to investigate effects of MCT1 silencing on expression of type 1 collagen, a major component of fibrosis. Cells were transfected with Lipofectamine and native MCT1-targeting siRNA (MCT1-siRNA), which diminished *SLC16A1/MCT1* mRNA expression by about 80% (*Figure 1A*), or nontargeted control (NTC-siRNA), and then treated with transforming growth factor 1β (TGF-β1) (10 μg/ml) for 48 hr. As expected, TGF-β1 stimulated expression of *ACTA2* and collagen 1 isoform, *COL1A1*, by several folds (*Figure 1B and C*). *SLC16A1/MCT1* silencing significantly inhibited TGF-β1-stimulated *ACTA2* mRNA expression as well as collagen 1 protein production (*Figure 1B and C*), indicating cell-autonomous functions of MCT1 in hepatic stellate cells.

### Identification of a potent, chemically modified siRNA candidate targeting MCT1

Given the therapeutic potential of *SLC16A1/MCT1* silencing in preventing fibrogenesis (*Figure 1*), we aimed to develop MCT1-siRNA compounds, chemically modified for stability, potency, and delivery in vivo for use in this research and potentially for therapeutic advancement. Asymmetrical siRNA compounds used here are composed of 15 double-strand nucleotides with a short overhanging single-strand that promotes cellular uptake (*Behlke, 2006*; *Khvorova and Watts, 2017*). To enhance the stability of the constructs, the 2′-OH of each ribose was modified to either 2′-*O*-methyl or 2′-fluoro. In addition, phosphorothioate linkage backbone modifications were applied to avoid exonuclease degradation. Tetra-ethylenglycol-cholesterol (Chol) was conjugated to the 3′ end-sense strand to enhance stability and cellular uptake of candidate compounds. Each Chol-conjugated, fully chemically modified MCT1-siRNA (Chol-MCT1-siRNA) candidate construct's sequence and targeting region on the *Slc16a1/Mct1* transcript is described in *Table 1* and *Figure 2A*.

We performed in vitro screening to select the most potent Chol-MCT1-siRNA compounds that were initially synthesized (*Figure 2A and B*). Each Chol-MCT1-siRNA compound candidate was treated into mouse hepatocyte FL83B cells. As opposed to native siRNA, our Chol-MCT1-siRNA does not require transfection reagents as it is fully chemically modified. The silencing effect on *Slc16a1/Mct1* mRNA was monitored after 72 hr (*Figure 2B*). Several compounds elicited a silencing effect greater than 80% compared to the Chol-NTC-siRNA. The two most potent Chol-MCT1-siRNA, Chol-MCT1-2060 (IC50: 59.6 nM, KD%: 87.2) and Chol-MCT1-3160 (IC50: 32.4 nM, KD%: 87.7) (*Figure 2C*), were evaluated for their inhibitory effect on MCT1 protein levels (*Figure 2D and E*). Based on its IC50 value and silencing potency, Chol-MCT1-3160 construct was chosen for further studies in vivo (*Table 2*).

### Distinct cellular biodistribution of Chol- vs GN-conjugated siRNAs

For in vivo studies, further chemical modifications were applied (*Figure 3A*). MCT1-siRNAs utilized in in vivo studies are double-strand oligonucleotides comprised of 18 sense and 20 antisense nucleotides. At the 5′-end of the antisense strand, a 5′-(*E*)-vinyl-phosphonate modification was added to

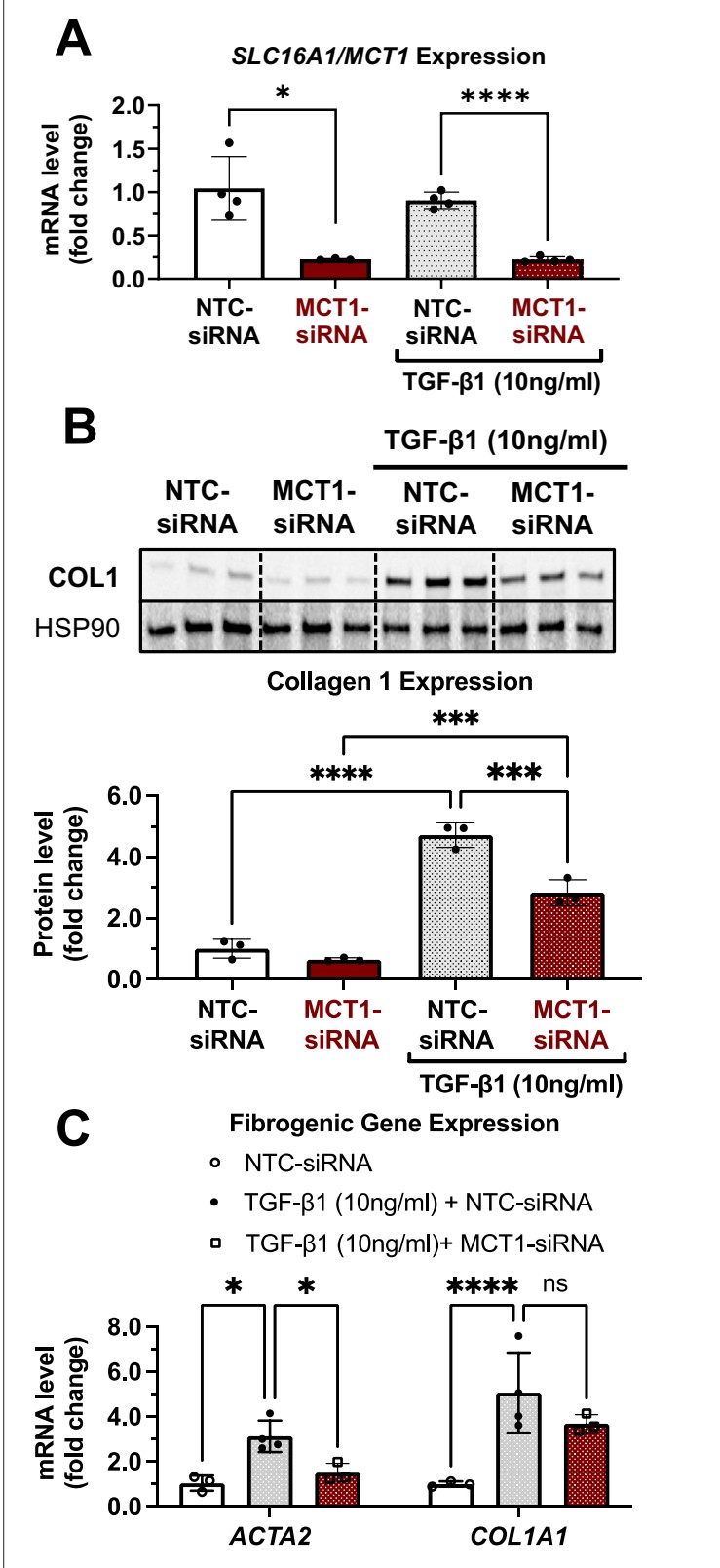

**Figure 1.** MCT1 depletion attenuates transforming growth factor 1β (TGF-β1)-stimulated collagen 1 production in human LX2 stellate cells. Cells were transfected with either NTC-siRNA or MCT1-siRNA for 6 hr. Then, cells were maintained in serum-starved media with or without 10 ng/ml of recombinant human TGF-β1 for 48 hr and harvested. (**A**) *SLC16A1*/MCT1 mRNA expression levels. (**B**) Collagen 1 protein levels. Quantification was added

*Figure 1 continued on next page*

*Figure 1 continued*

below. (**C**) Representative fibrogenic marker genes, *ACTA2, and COL1A1* expression levels were monitored (mean ± SD, t-test, one-way ANOVA, *: $p<0.05$, **: $p<0.01$, ***: $p<0.001$, ****: $p<0.0001$).

The online version of this article includes the following source data for figure 1:

**Source data 1.** MCT1 depletion attenuates transforming growth factor 1β (TGF-β1)-stimulated collagen 1 production in human LX2 stellate cells.

---

prevent phosphatase-induced degradation, enhancing in vivo stability and promoting its accumulation in target cells. Either a hydrophobic Chol or a hepatocyte-targeting GN was attached to the 3'-end of sense strands of MCT1-siRNAs to direct different hepatic cellular biodistribution.

To validate the biodistribution of siRNAs with these two conjugates, 10 mg/kg of each siRNA was subcutaneously injected into 16- to 18-week-old male C57BL/6 wild-type mice twice within a 15-day period. On day 15, mice were sacrificed and the livers were perfused to isolate multiple hepatic cell types, including hepatocytes, stellate cells, and Kupffer cells. Isolation of each hepatic cell type was validated for enrichment (*Figure 3—figure supplement 1A–F*). As expected, GN-conjugated, fully chemically modified MCT1-siRNA (GN-MCT1-siRNA) silenced *Slc16a1/Mct1* mRNA only in the hepatocyte fraction (*Figure 3B–D*), as GN binds to the asialoglycoprotein receptor primarily

**Table 1.** Sequences of chemically modified siRNA candidates targeting MCT1 used in in vitro screening.
siRNAs utilized in in vitro screening were a double-strand oligonucleotide comprised of 15 sense and 20 antisense nucleotides. The sequences of each candidate's antisense and sense strands were listed (P: 5'-phosphate, #: phosphorothioate, m: 2'-O-methyl, f: 2'-fluoro, Chol: tetra-ethylenglycol-cholesterol conjugate).

**Antisense strands:**

| Oligo ID | Chemically modified RNA sequence |
|---|---|
| MCT1-507 | P(mU)#(fG)#(mU)(fU)(mA)(fC)(mA)(fG)(mA)(fA)(mA)(fG)(mA)#(fA)#(mG)#(fC)#(mU)#(fG)#(mC)#(fG) |
| MCT1-1976 | P(mU)#(fA)#(mA)(fA)(mC)(fU)(mU)(fA)(mA)(fG)(mG)(fC)(mA)#(fC)#(mA)#(fU)#(mA)#(fU)#(mU)#(fA) |
| MCT1-2013 | P(mU)#(fU)#(mU)(fA)(mA)(fA)(mA)(fG)(mU)(fU)(mA)(fA)(mG)#(fG)#(mC)#(fU)#(mC)#(fU)#(mC)#(fU) |
| MCT1-2042 | P(mU)#(fU)#(mU)(fA)(mA)(fA)(mA)(fC)(mA)(fA)(mA)(fU)(mG)#(fA)#(mA)#(fU)#(mU)#(fC)#(mA)#(fG) |
| MCT1-2060 | P(mU)#(fU)#(mU)(fC)(mC)(fU)(mU)(fU)(mU)(fA)(mA)(fA)(mA)#(fU)#(mG)#(fA)#(mC)#(fA)#(mU)#(fU) |
| MCT1-2120 | P(mU)#(fU)#(mU)(fA)(mC)(fA)(mA)(fA)(mC)(fA)(mA)(fC)(mA)#(fA)#(mC)#(fA)#(mA)#(fA)#(mA)#(fC) |
| MCT1-3067 | P(mU)#(fU)#(mU)(fU)(mC)(fU)(mG)(fC)(mC)(fU)(mC)(fU)(mA)#(fU)#(mU)#(fC)#(mA)#(fG)#(mA)#(fA) |
| MCT1-3160 | P(mU)#(fU)#(mC)(fU)(mU)(fA)(mC)(fA)(mC)(fA)(mA)(fG)(mG)#(fU)#(mU)#(fU)#(mU)#(fA)#(mA)#(fA) |
| MCT1-3290 | P(mU)#(fA)#(mU)(fA)(mU)(fU)(mA)(fG)(mA)(fA)(mA)(fG)(mG)#(fU)#(mU)#(fA)#(mA)#(fA)#(mA)#(fU) |
| MCT1-4340 | P(mU)#(fU)#(mG)(fA)(mA)(fU)(mU)(fU)(mG)(fU)(mA)(fU)(mG)#(fA)#(mG)#(fA)#(mA)#(fU)#(mA)#(fA) |

**Sense strands:**

| Oligo ID | Chemically modified RNA sequence |
|---|---|
| Chol-MCT1-507 | (fC)#(mU)#(fU)(mC)(fU)(mU)(fU)(mC)(fU)(mG)(fU)(mA)(fA)#(mC)#(fA)-Chol |
| Chol-MCT1-1976 | (fU)#(mG)#(fU)(mG)(fC)(mC)(fU)(mU)(fA)(mA)(fG)(mU)(fU)#(mU)#(fA)-Chol |
| Chol-MCT1-2013 | (fG)#(mC)#(fC)(mU)(fU)(mA)(fA)(mC)(fU)(mU)(fU)(mU)(fA)#(mA)#(fA)-Chol |
| Chol-MCT1-2042 | (fU)#(mU)#(fC)(mA)(fU)(mU)(fU)(mG)(fU)(mU)(fU)(mU)(fA)#(mA)#(fA)-Chol |
| Chol-MCT1-2060 | (fC)#(mA)#(fU)(mU)(fU)(mU)(fA)(mA)(fA)(mA)(fG)(mG)(fA)#(mA)#(fA)-Chol |
| Chol-MCT1-2120 | (fG)#(mU)#(fU)(mG)(fU)(mU)(fG)(mU)(fU)(mU)(fG)(mU)(fA)#(mA)#(fA)-Chol |
| Chol-MCT1-3067 | (fA)#(mA)#(fU)(mA)(fG)(mA)(fG)(mG)(fC)(mA)(fG)(mA)(fA)#(mA)#(fA)-Chol |
| Chol-MCT1-3160 | (fA)#(mA)#(fC)(mC)(fU)(mU)(fG)(mU)(fG)(mU)(fA)(mA)(fG)#(mA)#(fA)-Chol |
| Chol-MCT1-3290 | (fA)#(mA)#(fC)(mC)(fU)(mU)(fC)(mU)(fU)(mA)(fA)(mU)(fA)#(mU)#(fA)-Chol |
| Chol-MCT1-4340 | (fC)#(mU)#(fC)(mA)(fU)(mA)(fC)(mA)(fA)(mA)(fU)(mU)(fC)#(mA)#(fA)-Chol |

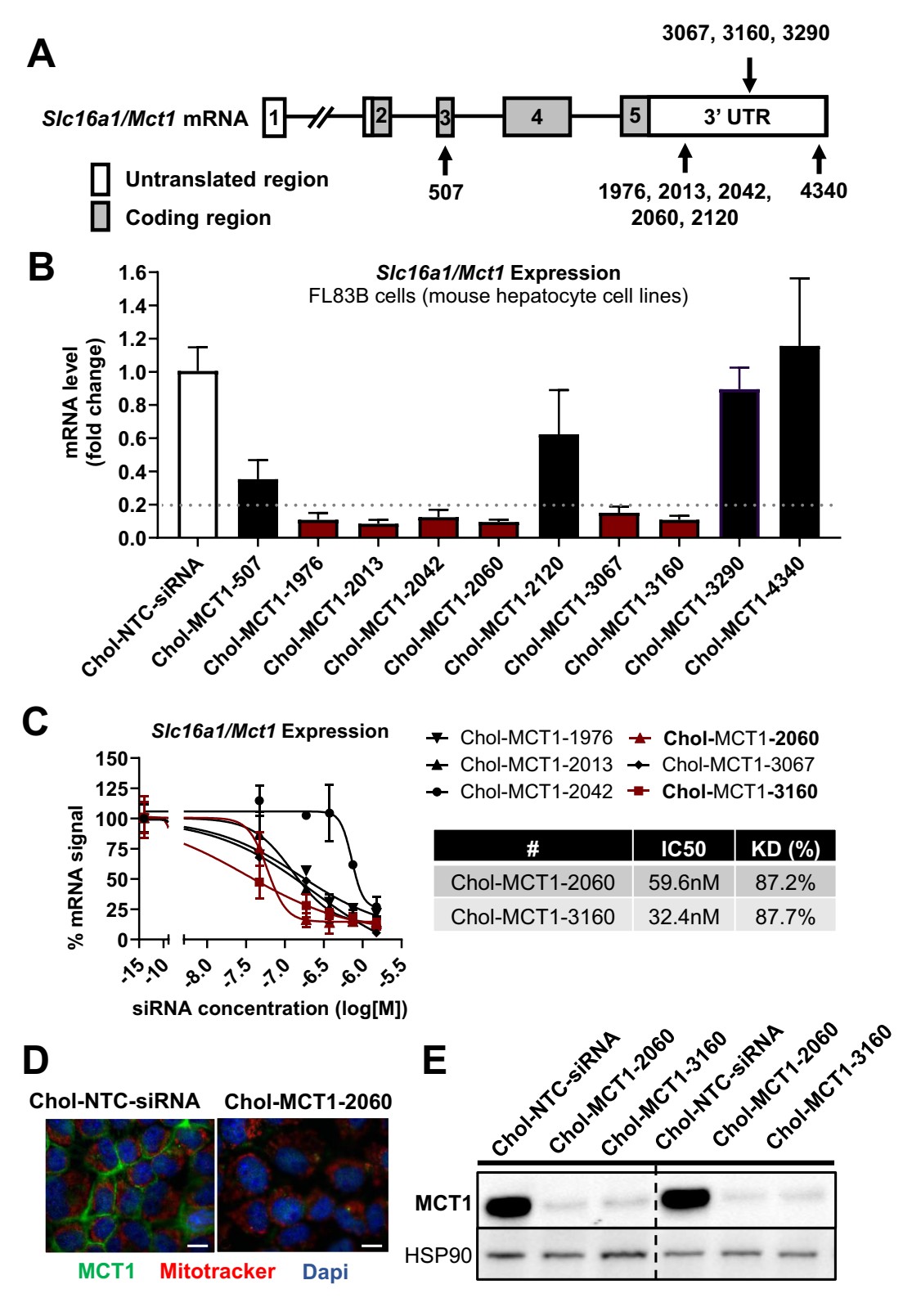

**Figure 2.** Screening of chemically modified Chol-MCT1-siRNA in vitro. (**A**) Targeted regions of multiple Chol-MCT1-siRNA candidates on *Slc16a1/Mct1* transcript. (**B**) Silencing efficacy of each Chol-MCT1-siRNA candidate (1.5 µM) on *Slc16a1/Mct1* mRNA expression levels was monitored 72 hr after the treatment in mouse hepatocyte cell lines, FL83B in vitro. Chol-NTC-siRNA was used as a control (mean ± SD). (**C**) Dose-response potency test was performed to identify the most potent Chol-MCT1-siRNA compound. IC50 values were determined using six serially diluted concentrations of

*Figure 2 continued on next page*

*Figure 2 continued*

each compound starting from 1.5 µM (mean ± SD). IC50 values and knockdown % of the two most potent compounds were shown in the table below. (D) 72 hr after the treatment of Chol-MCT1-2060 compounds (1.5 µM), MCT1 protein expression levels were visually monitored by immunofluorescence (scale bar: 10 µm). (E) 72 hr after the treatment of either Chol-MCT1-2060 or Chol-MCT1-3160 compounds (1.5 µM), their silencing efficacy on MCT1 protein expression levels was examined by western blotting.

The online version of this article includes the following source data for figure 2:

**Source data 1.** Screening of chemically modified Chol-MCT1-siRNA in vitro.

expressed in hepatocytes. On the other hand, Chol-MCT1-siRNA silenced *Slc16a1/Mct1* mRNA levels in all hepatic cell types (*Figure 3E–G*), as its cellular uptake is highly dependent on the non-specific, hydrophobic interaction between cholesterol and plasma membranes. Notably, the hepatic stellate cell fraction distinguishes GN-MCT1-siRNA from Chol-MCT1-siRNA in biodistribution, as only the latter silences *Slc16a1/Mct1* in stellate cells (*Figure 3C vs F*). We also confirmed that both GN-MCT1-siRNA and Chol-MCT1-siRNA do not affect MCT1 levels in other major metabolic tissues (*Figure 3—figure supplement 1G–J*).

## Subcutaneous injection of Chol-MCT1-siRNA or GN-MCT1-siRNA silences hepatic MCT1 in a genetically obese NASH mouse model

We next investigated the effect of Chol-MCT1-siRNA on reversing severe steatosis in the genetically obese ob/ob mouse on a NASH-inducing Gubra Amylin NASH (GAN) diet (*Yenilmez et al., 2022*). These mice normally develop severe steatosis from an early age but hardly develop fibrosis until fed the GAN diet. Each siRNA (10 mg/kg) was subcutaneously injected once every 10 days and mice were fed a GAN diet for 3 weeks before sacrifice (*Figure 4A*). Hepatic MCT1 protein levels were visually monitored by MCT1-positive staining immunohistochemistry (*Figure 4B and C*), showing more than 70% MCT1 protein depletion (Chol-MCT1-siRNA: 77.99% and GN-MCT1-siRNA: 71.35% silencing). Similar silencing was observed when *Slc16a1/Mct1* mRNA levels were measured by real-time quantitative PCR (rt-qPCR) (*Figure 4D*). The silencing was *Slc16a1/Mct1* selective, not depleting other isoforms such as *Slc16a7/Mct2* and *Slc16a3/Mct4* (*Figure 4—figure supplement 1A and B*). Importantly, there was no surge in plasma lactate level (*Figure 4E*), addressing the concern of potential lactic acidosis after MCT1 depletion in the liver, the major lactate-consuming tissue. We also monitored food intake and body weight over time (*Figure 4—figure supplement 1C and D*), as there was a report of decreased food anticipation activity upon hepatic MCT1 deletion followed by reduced plasma β-hydroxybutyrate levels (*Martini et al., 2021*). Intriguingly, the GN-MCT1-siRNA administration led to a decrease in both food intake and body weight, while the Chol-MCT1-siRNA did not. Neither Chol-MCT1-siRNA administration nor hepatocyte-specific MCT1KO improved glucose tolerance on the genetically obese NASH mouse model or a 12-week HFD-induced NAFLD model, respectively (*Figure 4—figure supplement 1E and F*).

**Table 2.** Sequences of the selected final chemically modified siRNA candidates targeting MCT1 used for in vivo studies. MCT1-3160 was selected for the final construct for in vivo studies. MCT1-siRNAs utilized in in vivo study was a double-strand oligonucleotide comprised of 18 sense and 20 antisense nucleotides. To sense strands, either Chol- or GN- was attached (VP: 5'-(E)-vinyl phosphonate, #: phosphorothioate, m: 2'-O-methyl, f: 2'-fluoro, Chol: tetra-ethylenglycol-cholesterol conjugate, GN: tri-N-acetyl-galactosamine).

**Antisense strands:**

| Oligo ID | Chemically modified RNA sequence |
|---|---|
| MCT1-3160 | VP(mU)#(fU)#(mC)(mU)(mU)(fA)(mC)(mA)(mC)(mA)(mA)(mG)(mG)#(fU)#(mU)#(fU)#(mU)#(mA)#(mA)#(fA) |

**Sense strands:**

| Oligo ID | Chemically modified RNA sequence |
|---|---|
| Chol-MCT1-3160 | (mU)#(mA)#(mA)(mA)(mA)(mC)(mC)(fU)(fU)(fG)(mU)(fG)(mU)(mA)(mA)(mG)#(mA)#(mA)-Chol |
| GN-MCT-3160 | (mU)#(mA)#(mA)(mA)(mA)(mC)(mC)(fU)(fU)(fG)(mU)(fG)(mU)(mA)(mA)(mG)#(mA)#(mA)-GN |

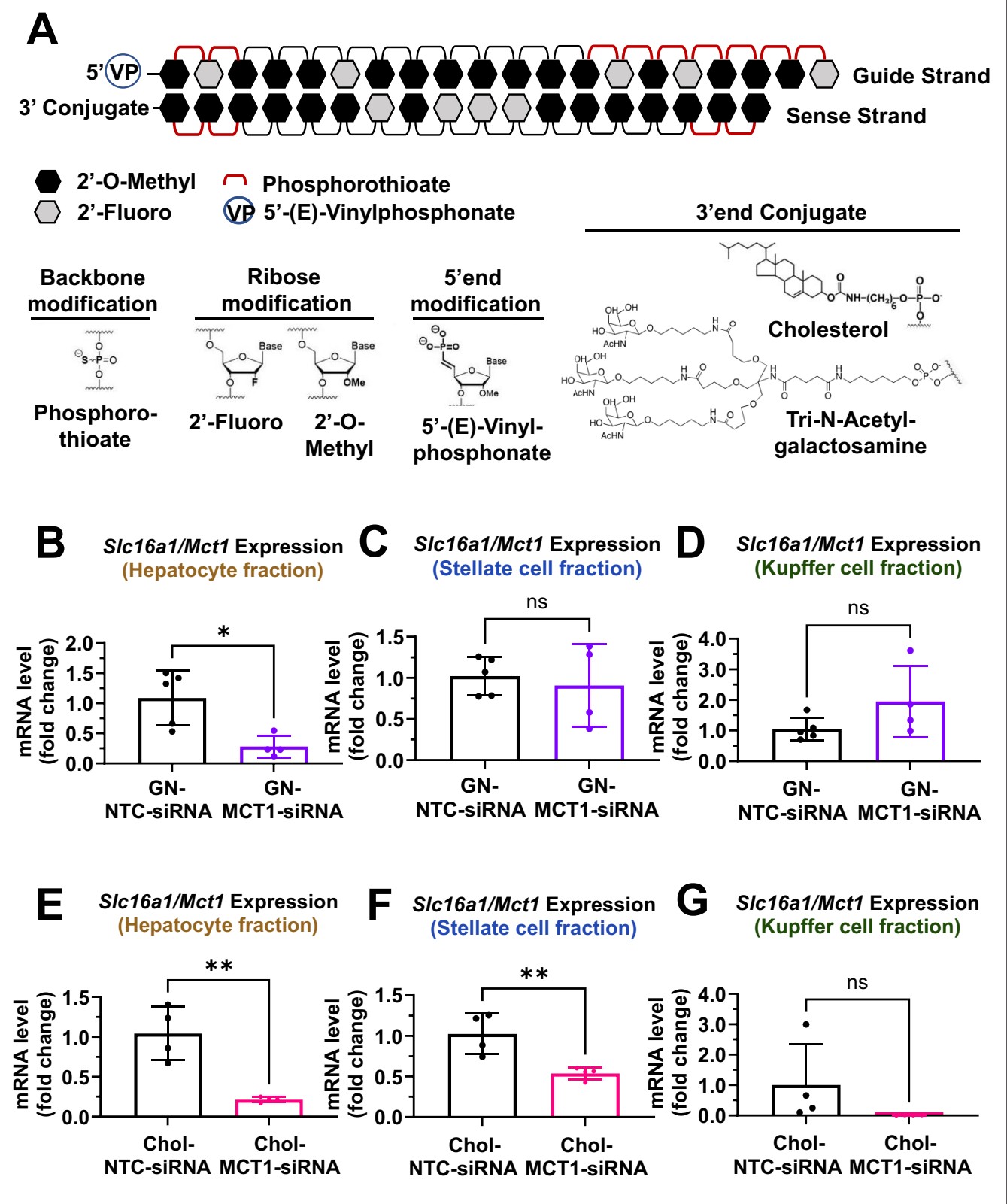

**Figure 3.** Biodistribution of Chol- and GN-MCT1-siRNA in the liver. Male C57BL/6 wild-type mice (16–18 weeks, n=4) were subcutaneously injected with 10 mg/kg of each siRNA, twice within 15 days, while fed a chow diet. Mice were sacrificed on day 15. (**A**) Chemical structure of the fully chemically modified siRNA that was used for further in vivo studies: Chol-MCT1-siRNA and GN-MCT1-siRNA. (**B, E**) Primary hepatocytes, (**C, F**) stellate cells, and

*Figure 3 continued on next page*

*Figure 3 continued*

(D, G) Kupffer cells were isolated from each mouse using different gravity centrifugations and gradient solutions after the liver perfusion. *Slc16a1/Mct1* mRNA expression levels in each cell-type fraction were measured (mean ± SD, t-test, *: $p<0.05$, **: $p<0.01$).

The online version of this article includes the following source data and figure supplement(s) for figure 3:

**Source data 1.** Biodistribution of Chol- and GN-MCT1-siRNA in the liver.

**Figure supplement 1.** Biodistribution of GN-MCT1-siRNA and Chol-MCT1-siRNA.

**Figure supplement 1—source data 1.** Biodistribution of GN-MCT1-siRNA and Chol-MCT1-siRNA.

## Hepatic MCT1 depletion downregulates lipogenic genes but not steatosis in the ob/ob NASH diet mouse model

In order to fully analyze steatosis in NASH, lipid droplet morphology and total hepatic TG were assessed (*Figure 4F–H*). The results showed that neither GN-MCT1-siRNA nor Chol-MCT1-siRNA decreased total hepatic TG levels (*Figure 4H*), although quantitative analysis of H&E images showed a small decrease in mean lipid droplet size and increased number of lipid droplets upon MCT1 silencing (*Figure 4F and G*). These data suggest the possibility that hepatic MCT1 depletion either (1) inhibits formation or fusion of lipid droplets, or (2) enhances lipolysis to diminish lipid droplet size.

To investigate the underlying mechanism by which lipid droplet morphological dynamics change, we monitored the effect of hepatic MCT1 depletion on DNL-related gene expression. Both GN-MCT1-siRNA and Chol-MCT1-siRNA strongly decreased the mRNA and protein levels related to representative DNL genes (*Figure 4—figure supplement 2A–D*). Intriguingly, both modes of hepatic MCT1 depletion also inhibited expression of the upstream regulatory transcription factors SREBP1 and ChREBP. Because phosphorylated AMPK (pAMPK), an active form of AMPK, is known to inhibit SREBP nuclear translocation (*Li et al., 2011*) as well as the DNA binding activity of ChREBP (*Kawaguchi et al., 2002*), we evaluated pAMPK levels. As a result, there was a significant increase in pAMPK levels and pAMPK/AMPK ratio in both GN-MCT1-siRNA and Chol-MCT1-siRNA injected groups (*Figure 4—figure supplement 2E and F*).

## Opposite effects of Chol-MCT1-siRNA versus GN-MCT1-siRNA on fibrotic collagen expression

We next monitored fibrosis, a tissue damaging phenotype that is associated with NASH. Consistent with the results in LX2 stellate cells (*Figure 1*), Chol-MCT1-siRNA administration to ob/ob mice on GAN diet significantly reduced liver collagen 1 protein levels (*Figure 5A and B*). This result could be attributable to the fact that subcutaneous injection of the Chol-MCT1-siRNA compound is able to silence genes in hepatic stellate cells, the predominant cell type that produces collagens. Interestingly, decreases in mRNA encoding collagen 1 isoforms were not detected by rt-qPCR analysis (*Figure 5C*), indicating possible effects at the level of translation or protein turnover. Surprisingly, an opposite phenotype on collagen 1 protein expression was observed in response to administration of GN-MCT1-siRNA compared to Chol-MCT1-siRNA (*Figure 5D and E*). Hepatocyte-specific GN-MCT1-siRNA actually enhanced the expression of type 1 collagen protein in these experiments (*Figure 5D and E*), and this effect was also apparent at the mRNA expression level (*Figure 5F*). Overall fibrosis as detected by Sirius Red was also analyzed in these experiments, as this staining detects all types of collagen fibers that are involved in hepatic fibrosis, such as III, IV, V, and VI. In line with collagen 1 mRNA and protein levels, GN-MCT1-siRNA significantly enhanced Sirius Red positive areas in the images, as shown in *Figure 5G and H*. Despite its inhibitory effect on collagen 1 production levels, Chol-MCT1-siRNA did not reduce Sirius Red positive areas.

## A comparable level of M1/M2 macrophage polarization upon GN-MCT1-siRNA and Chol-MCT1-siRNA administration

Given the distinct hepatic cellular distribution of Chol-MCT1-siRNA and GN-MCT1-siRNA (*Figure 3—figure supplement 1*), the opposite fibrogenic phenotype observed may be attributed to MCT1's role in non-hepatocyte cell types such as the inflammatory Kupffer cells and the fibrogenic hepatic stellate cells. To determine which hepatic cell type derived the opposite fibrotic phenotypes of MCT1, we first hypothesized that GN-MCT1-siRNA activates M2 pro-fibrogenic macrophage more than

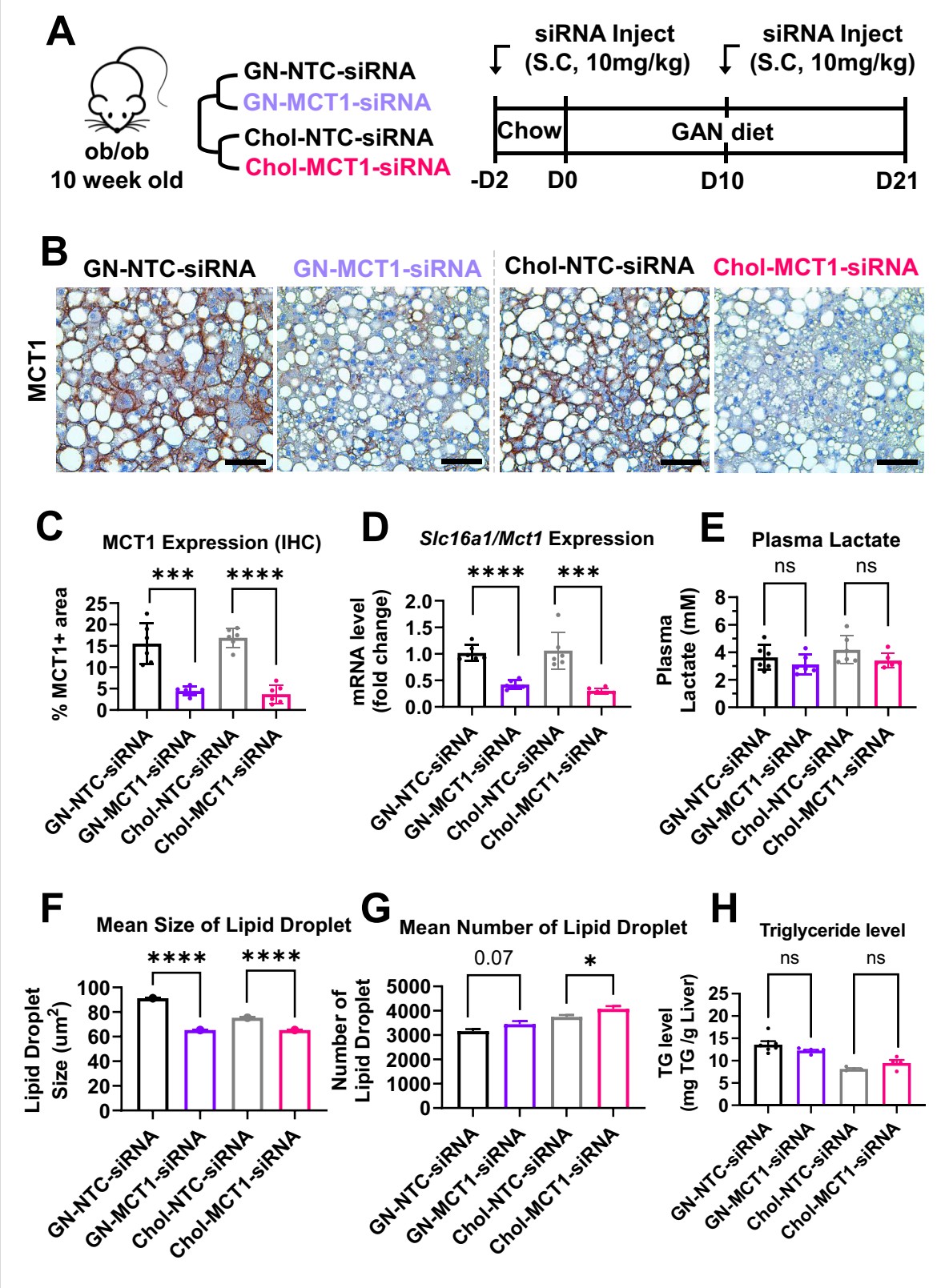

**Figure 4.** Hepatic MCT1 depletion did not resolve steatosis in a genetically obese nonalcoholic steatohepatitis (NASH) mouse model. (**A**) Male ob/ob mice (10 weeks, n=6) were subcutaneously injected with 10 mg/kg of siRNA once every 10 days. Mice were fed a Gubra Amylin NASH (GAN) diet for 3 weeks and sacrificed. (**B**) Livers were stained with MCT1 antibody and the representative images of each group are shown (scale bar: 50 μm). (**C**) % of MCT1 positive area shown in immunohistochemistry images were quantified. (**D**) Hepatic *Slc16a1/Mct1* mRNA level was measured by real-time

*Figure 4 continued on next page*

*Figure 4 continued*

quantitative PCR (rt-qPCR) upon each siRNA administration. (**E**) Plasma lactate levels were monitored. (**F**) Mean size of lipid droplets was quantified from H&E images (mean, sem). (**G**) Mean number of lipid droplets was quantified from H&E images (mean, sem). (**H**) Liver triglyceride (TG) levels were examined in each group (mean ± SD or otherwise noted, t-test, *: p<0.05, ***: p<0.001, ****: p<0.0001).

The online version of this article includes the following source data, source code, and figure supplement(s) for figure 4:

**Source code 1.** Hepatic MCT1 depletion did not resolve steatosis in a genetically obese nonalcoholic steatohepatitis (NASH) mouse model.

**Source data 1.** Hepatic MCT1 depletion did not resolve steatosis in a genetically obese nonalcoholic steatohepatitis (NASH) mouse model.

**Figure supplement 1.** GN-MCT1-siRNA induced a complementary effect on monocarboxylate transporter (MCT) isoform expression and decreased food intake and body weight.

**Figure supplement 1—source data 1.** GN-MCT1-siRNA induced a complementary effect on monocarboxylate transporter (MCT) isoform expression and decreased food intake and body weight.

**Figure supplement 2.** Both Chol-MCT1-siRNA and GN-MCT1-siRNA significantly decreased hepatic DNL gene expression.

**Figure supplement 2—source data 1.** Both Chol-MCT1-siRNA and GN-MCT1-siRNA significantly decreased hepatic DNL gene expression.

Chol-MCT1-siRNA does. Since Zhang et al. have demonstrated that histone lactylation via lactyl-CoA intermediate drives the shift of M1 macrophages into pro-fibrogenic M2-like macrophage polarization upon bacterial exposure, the epigenetic contribution of lactate via lactylation has started to be appreciated (*Zhang et al., 2019*; *Liu et al., 2022*; *Gaffney et al., 2020*). Similarly, Cui et al. have shown that histone lactylation promotes macrophage profibrotic activity via p300 in lung myofibroblasts (*Cui et al., 2021*). Interestingly, another report claims targeting MCT1-mediated lactate flux attenuates pulmonary fibrosis by preventing macrophage profibrotic polarization (*He et al., 2023*). These findings suggest the possibility that reduced histone lactylation in macrophages, indirectly affected by MCT1 depletion in the stellate cells, prevents fibrogenesis and inhibits pro-fibrogenic M2 macrophage polarization. Thus, representative M1 pro-inflammatory macrophage and M2 pro-fibrogenic macrophage markers were monitored in our experiments. However, GN-MCT1-siRNA treatment caused comparable M1/M2 macrophage activation levels to Chol-MCT1-siRNA treatment (*Figure 5—figure supplement 1A and B*). These data suggest that the opposite fibrotic phenotypes caused by the different siRNA constructs are not due to M1/M2 macrophage polarization.

## MCT1KO by AAV-Lrat-Cre and AAV-TBG-Cre constructs confirm cell-type specificity of MCT1KO effects

The results presented above suggested that Chol-MCT1-siRNA downregulates type 1 collagen protein by depleting MCT1 in hepatic stellate cells, which hepatocyte-specific GN-MCT1-siRNA cannot target. To test this hypothesis, we developed and validated AAV9-Lrat-Cre constructs to generate hepatic stellate cell-specific MCT1 knockout mice. Male MCT1[fl/fl] mice (*Jha et al., 2020*) were intravenously injected with AAV9-Lrat-Cre ($1 \times 10^{11}$ gc) and sacrificed 3 weeks later (*Figure 5—figure supplement 2A*). Isolation of hepatocytes and hepatic stellate cells was validated with their representative marker genes encoding albumin and desmin, respectively (*Figure 5—figure supplement 2B and C*). Successful depletion of hepatic stellate cell-selective *Slc16a1Mct1* mRNA was confirmed in the MCT1[fl/fl] mice injected with the AAV9-Lrat-Cre construct. *Slc16a1/Mct1* mRNA levels in the hepatocytes, which account for up to 70% of total liver cell types, were intact (*Figure 5—figure supplement 2D and E*). No change in MCT1 protein level was observed in other metabolic tissues (*Figure 5—figure supplement 2F and G*).

## Hepatocyte-specific MCT1KO accelerated fibrosis, while hepatic stellate cell-specific MCT1KO decreased it in the CDHFD-induced NASH mouse model

We employed the choline-deficient, high-fat diet (CDHFD)-induced NASH mouse model to test these AAV constructs on steatosis and fibrosis. CDHFD induces severe steatosis due to inhibited VLDL secretion and β-oxidation and thereby exacerbates NASH fibrosis in a relatively short time (*Raubenheimer et al., 2006*; *Matsumoto et al., 2013*). MCT1[fl/fl] mice were intravenously injected with $2 \times 10^{11}$ gc of AAV8-TBG-Cre or AAV9-Lrat-Cre or both (*Figure 6A*). A week after the injections, mice were fed with a CDHFD for 8 weeks to induce NASH. Depletion of hepatic *Slc16a1/Mct1* mRNA in each group

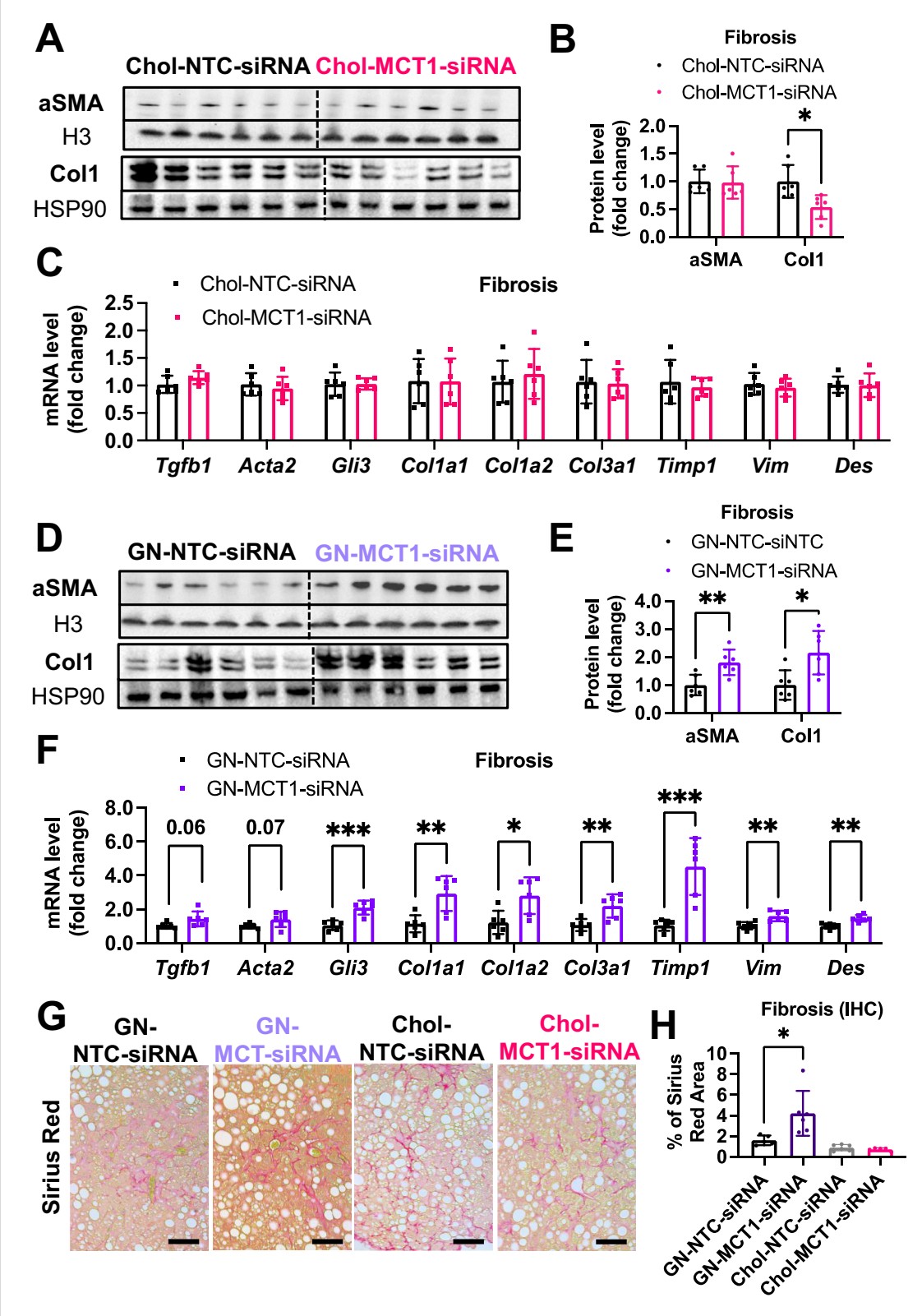

**Figure 5.** Opposite effects of Chol-MCT1-siRNA versus GN-MCT1-siRNA on fibrotic type 1 collagen expression. Male ob/ob mice (10weeks, n=6) were subcutaneously injected with 10mg/kg of siRNA once every 10days. Mice were fed a Gubra Amylin NASH (GAN) diet for 3weeks and sacrificed. Representative fibrogenic gene expression levels were measured for (**A, B**) mRNA and (**C, D**) protein. (**E, F**) Protein expression levels were quantified.

*Figure 5 continued on next page*

*Figure 5 continued*

(**G**) Livers were stained with Sirius Red and the representative images of each group are shown (scale bar: 200 µm). (**H**) % of Sirius Red positive areas were quantified (mean ± SD, t-test, *: p<0.05, **: p<0.01, ***: p<0.001).

The online version of this article includes the following source data, source code, and figure supplement(s) for figure 5:

**Source code 1.** Opposite effects of Chol-MCT1-siRNA versus GN-MCT1-siRNA on fibrotic type 1 collagen expression.

**Source data 1.** Opposite effects of Chol-MCT1-siRNA versus GN-MCT1-siRNA on fibrotic type 1 collagen expression.

**Figure supplement 1.** A comparable level of M1/M2 macrophage polarization upon Chol-MCT1-siRNA and GN-MCT1-siRNA administration.

**Figure supplement 1—source data 1.** A comparable level of M1/M2 macrophage polarization upon Chol-MCT1-siRNA and GN-MCT1-siRNA administration.

**Figure supplement 2.** Intravenous injection of AAV9-Lrat-Cre in MCT1[fl/fl] mice specifically targets hepatic stellate cells.

**Figure supplement 2—source data 1.** Intravenous injection of AAV9-Lrat-Cre in MCT1[fl/fl] mice specifically targets hepatic stellate cells.

was confirmed (*Figure 6B*). There was no food intake or body weight difference between the groups (*Figure 6C and D*). Similar to the results we obtained by MCT1 silencing with siRNAs (*Figure 4*), MCT1 deletion in either hepatocytes or in hepatic stellate cells did not resolve steatosis (*Figure 6E and F*).

Also consistent with the results obtained by hepatocyte selective MCT1 silencing with GN-MCT1-siRNA (*Figure 5*), hepatocyte-specific knockout of MCT1 (Hep KO) enhanced the collagen 1 level compared to the control group (*Figure 7A*). In contrast, hepatic stellate cell-specific MCT1 knockout (HSC KO) prevented CDHFD-induced collagen 1 protein levels (*Figure 7B*). MCT1KO in combined hepatocytes and hepatic stellate cells blunted the effect shown in each single KO (*Figure 7C*). Overall liver fibrosis detected by trichrome staining again confirmed the acceleration of fibrosis in the Hep KO group and a downward trend in the HSC KO group (*Figure 7D and E*). Dual MCT1KO in hepatocytes plus hepatic stellate cells showed no change in overall fibrosis, similar to the Chol-MCT1-siRNA results (*Figure 5G and H*). Additionally, liver stiffness was monitored via ultrasound-based shear wave elastography (SWE) in a noninvasive diagnostic mode for liver disease (*Morin et al., 2021*; *Czernusze-wicz et al., 2022*). After 8 weeks of CDHFD, all groups had the same level of increased liver stiffness above what the control mice showed at 4 weeks, however, Hep KO mice exhibited elevated liver stiffness over all other groups at 4 weeks of the diet (*Figure 7F and G*). There was no change in plasma alanine transaminase (ALT) levels among the groups (*Figure 7H*).

## Lactate enhances the TGF-β1-stimulatory effect in the presence of pyruvate

To understand the underlying pathway explaining the opposite effects of GN-MCT1-siRNA and Chol-MCT1-siRNA on fibrotic collagen expression, we hypothesize that the lactate transporter MCT1 in hepatic stellate cells promotes the expression of fibrotic collagens by regulating lactate flux. Furthermore, we propose that in the case of MCT1 depletion in hepatocytes, lactate present in the hepatic blood flow that has not been taken up by hepatocytes is redirected to other hepatic cell types, including hepatic stellate cells. To test this hypothesis, we examined whether lactate itself acts as a fibrogenic inducer in cultured LX2 cells treated with increasing doses of sodium lactate (NaLac: 0, 2.5, 5, 10, 20, 40 mM) for 48 hr and then harvested. However, no increase in collagen mRNA or protein levels upon lactate treatment was observed (*Figure 8A and B*), suggesting that lactate itself may not be a fibrogenic inducer. We then investigated whether lactate assists in collagen production in the presence of other potent fibrogenic inducers, such as TGF-β1. LX2 cells were treated with three different conditions for 48 hr: (1) TGF-β1 only, (2) TGF-β1 with sodium pyruvate (NaPyr: 1 mM), and (3) TGF-β1 with the combination of sodium pyruvate (1 mM) and sodium lactate (10 mM) matched to the physiological lactate:pyruvate (L:P) ratio. Interestingly, the combination treatment of sodium pyruvate (1 mM) and sodium lactate (10 mM) significantly enhanced both mRNA and protein expressions of collagen 1 (*Figure 8C–E*). As the cells were grown in the high glucose (25 mM) DMEM condition, we further tested if MCT1 depletion can inhibit both endogenous and exogenous lactate-mediated collagen 1 production. MCT1 depletion in LX2 cells prevented TGF-β1-stimulated collagen 1 production in both conditions where lactate was solely generated by endogenous glycolysis (*Figure 8F*) and where exogenous lactate was supplied (*Figure 8G*). Taken together, these findings suggest that although lactate itself is not a fibrogenic inducer, it can assist in TGF-β1-induced collagen production via MCT1.

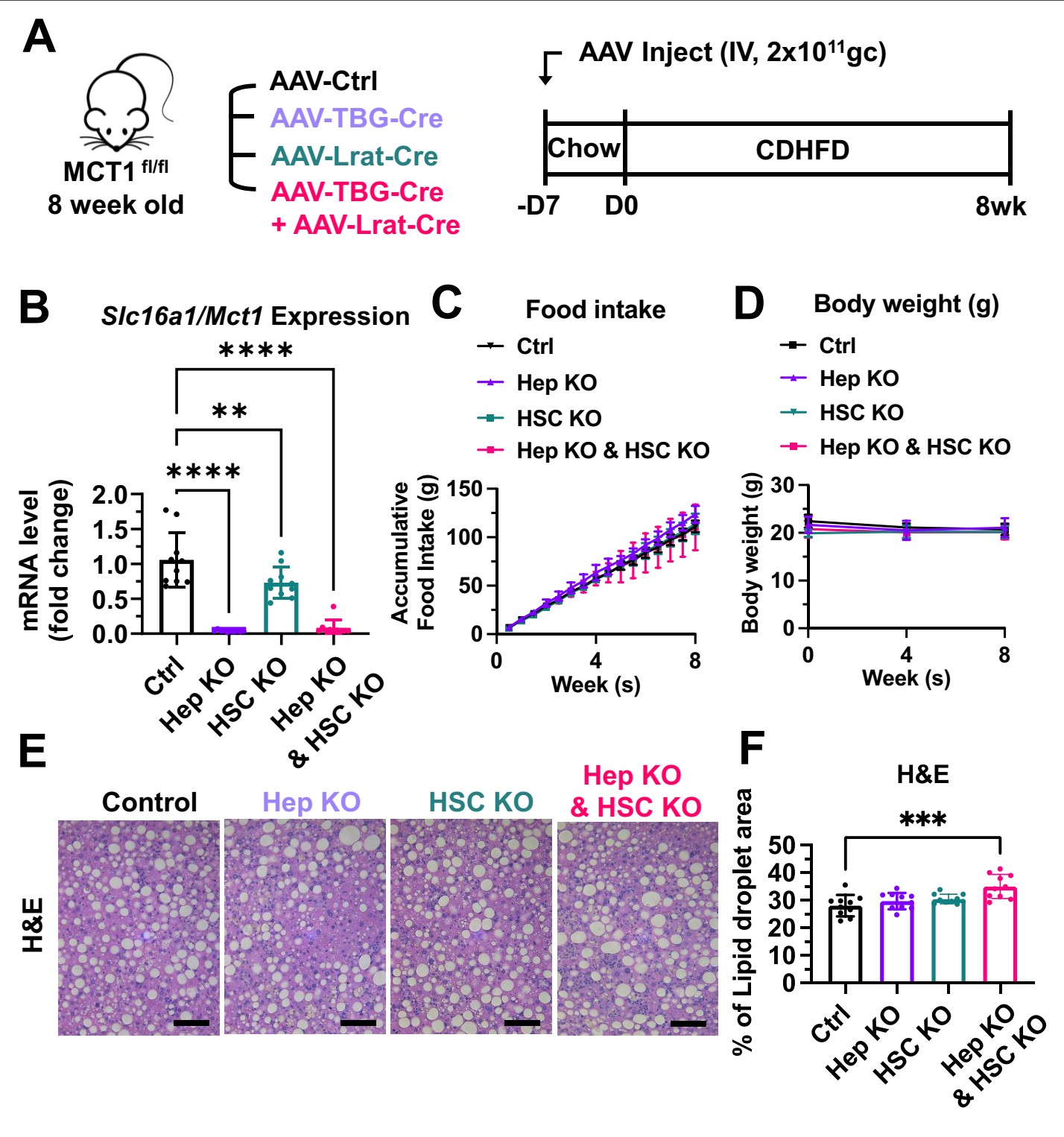

**Figure 6.** MCT1 depletion did not resolve steatosis in the choline-deficient, high-fat diet (CDHFD)-induced nonalcoholic steatohepatitis (NASH) model. (**A**) Male MCT1[fl/fl] mice (8 weeks, n=10) were intravenously injected with $2 \times 10^{11}$ gc of AAV-TBG-Cre or AAV-Lrat-Cre or both. The same amount of AAV-TBG-null or AAV-Lrat-null was used as a control. A week after the injection, mice were fed a CDHFD for 8 weeks and sacrificed. (**B**) *Slc16a1/Mct1* mRNA expression levels in whole livers were examined. (**C**) Food intake and (**D**) body weights were monitored. (**E**) CDHFD-induced steatosis was monitored by H&E (scale bar: 200 μm). (**F**) % of lipid droplet areas was quantified (mean ± SD, one-way ANOVA, *: $p<0.05$, **: $p<0.01$, ***: $p<0.001$, ****: $p<0.0001$).

The online version of this article includes the following source data for figure 6:

*Figure 6 continued on next page*

*Figure 6 continued*

**Source code 1.** MCT1 depletion did not resolve steatosis in the choline-deficient, high-fat diet (CDHFD)-induced nonalcoholic steatohepatitis (NASH) model.

**Source data 1.** MCT1 depletion did not resolve steatosis in the choline-deficient, high-fat diet (CDHFD)-induced nonalcoholic steatohepatitis (NASH) model.

## MCT1 promotes SMAD3 phosphorylation/activation in LX2 cells

We also examined the phosphorylation of SMAD3 which is the canonical pathway of TGF-β1-induced collagen production (*Breitkopf et al., 2006*; *Friedman, 2008Tsuchida and Friedman, 2017*). As a result, we observed that MCT1-siRNA significantly decreased the phosphorylated SMAD3 (pSMAD3) protein level and the pSMAD3/SMAD3 ratio in the absence of TGF-β1 treatment (*Figure 8—figure supplement 1A and B*), implying the potential regulatory effect of MCT1 on the TGF-β1-SMAD3 axis. Interestingly, however, in the presence of TGF-β1 pretreatment, the contribution of pSMAD on MCT1's inhibitory effect on collagen 1 production was limited. While MCT1 depletion inhibited collagen production, the significant decrease of pSMAD3/SMAD ratio was only shown upon low concentrations of TGF-β1 (0, 1, 2.5 ng/ml) pretreatment (*Figure 8—figure supplement 1C–E*). This data suggests that both SMAD3-dependent and independent MCT1 regulation may be relevant to the TGF-β1-induced collagen production in which MAPK, NF-kB, and PI3K are potential mediators in SMAD3-independent TGF-β1 pathways (*Derynck and Zhang, 2003*; *Xu et al., 2016*). However, the mechanism of these pathways and biological consequences are still not fully understood.

## MCT1 depletion enhanced fibrogenic gene expression levels in human hepatoma HepG2 cells

The direct effect of MCT1 depletion in HepG2 hepatocytes was also investigated by transfection with either NTC-siRNA or MCT1-siRNA (*Figure 8—figure supplement 2A and B*). MCT1-siRNA treatment depleted *SLC16A1/MCT1* mRNA levels (*Figure 8—figure supplement 2A*) and significantly enhanced fibrogenic gene markers *ACTA2* and *COL1A1* (*Figure 8—figure supplement 2B*). However, HepG2-derived conditioned media-treated LX2 cells did not change the expression of fibrogenic genes (*Figure 8—figure supplement 2C*). These data are consistent with the idea that enhanced fibrogenesis due to GN-MCT1-siRNA treatment of mice may be a direct effect of hepatocyte MCT1 depletion. However, hepatic stellate cells are thought to be the primary producers of collagen, contributing 10–20 times more collagen than hepatocytes or endothelial cells, respectively, in rodent systems (*Friedman et al., 1985*). Thus, this issue will require additional work to fully understand the basis for the effect of GN-MCT1-siRNA treatment.

## Discussion

The major finding of this study is that MCT1 function in hepatic stellate cells promotes collagen 1 expression, as MCT1 depletion in this cell type, either in vitro in cell culture (*Figure 1B*) or in vivo in mice (*Figure 7B*), attenuates fibrotic collagen 1 protein production. We also provided evidence that lactate may enhance TGF-β1-induced collagen production via MCT1 in studies in cultured LX2 cells. The finding in mice was made by generating novel AAV9-Lrat-Cre constructs that can be injected into MCT1^fl/fl mice to elicit stellate cell-selective MCT1 depletion, as verified by isolation and *Slc16a1/Mct1* mRNA analysis in liver cell types (*Figure 5—figure supplement 2*). Previous use of Lrat-Cre for germline transmission had validated constitutive gene KO selectively in hepatic stellate cells in mice (*Mederacke et al., 2013*), while our AAV-Lrat-Cre construct allows inducible gene KO, eliminating time-demanding mouse crossing and breeding. Surprisingly, MCT1KO in hepatocytes, both in vivo (*Figure 7A*) and in vitro (*Figure 8—figure supplement 2B*), evoked the opposite effect: a robust upregulation of collagen 1 expression and fibrosis. This may be an effect of increased local lactate to enhance stellate cell collagen production, but may also be in part a cell-autonomous effect in hepatocytes, as we observed increased collagen 1 expression in Hep2G cells upon silencing *SLC16A1/MCT1* (*Figure 8—figure supplement 2*). The hepatocyte-selective MCT1KO caused increased total fibrosis, evidenced by trichrome staining of liver (*Figure 7D and E*), while the stellate cell MCT1KO was associated with a trend toward diminished trichrome staining that did not reach statistical significance

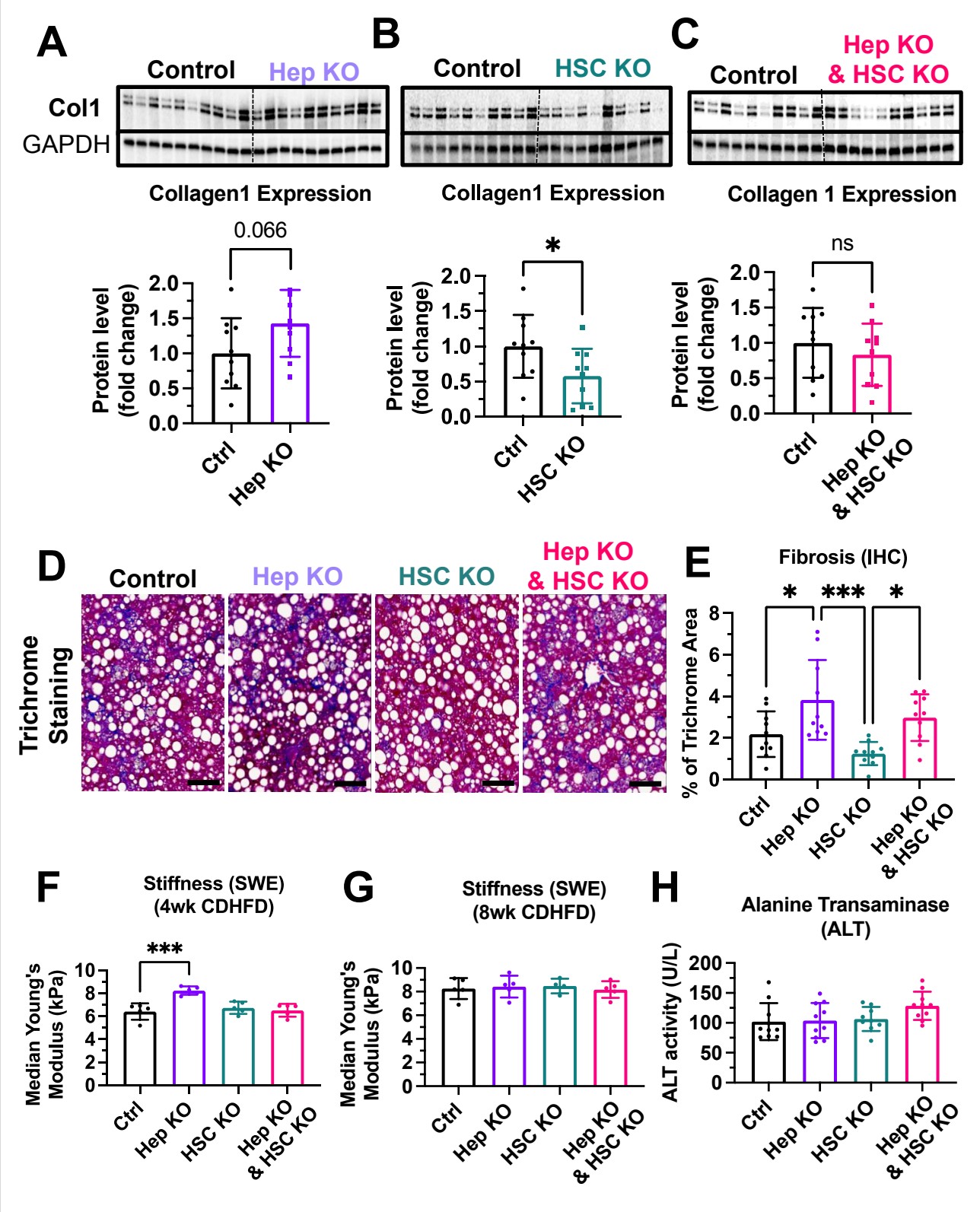

**Figure 7.** Hepatocyte-specific MCT1KO accelerated fibrosis, while hepatic stellate cell-specific MCT1KO decreased it. Male MCT1$^{fl/fl}$ mice (6 weeks, n=10) were intravenously injected with 2×10$^{11}$ gc of AAV-TBG-Cre or AAV-Lrat-Cre or both. The same amount of AAV-TBG-null or AAV-Lrat-null was used as a control. A week after the injection, mice were fed a choline-deficient, high-fat diet (CDHFD) for 8 weeks and sacrificed. (**A**) Collagen 1 protein levels were compared between the control and the hepatocyte MCT1KO groups. (**B**) Collagen 1 protein levels were compared between the control and

*Figure 7 continued on next page*

*Figure 7 continued*

the hepatic stellate cell MCT1KO groups. (**C**) Collagen 1 protein levels were compared between the control group and MCT1KO in both hepatocyte and hepatic stellate cell groups. (**D**) Livers were stained with trichrome and the representative images of each group were shown (scale bar: 100 µm). (**E**) Trichrome staining images were quantified. (**F**) Liver stiffness was monitored 4 weeks after CDHFD feeding via shear wave elastography (SWE). (**G**) Liver stiffness was monitored 8 weeks after CDHFD feeding via SWE. (**H**) Alanine transaminase (ALT) levels were measured in every CDHFD-fed group (mean ± SD, t-test, one-way ANOVA, *: $p<0.05$, **: $p<0.01$, ***: $p<0.001$, ****: $p<0.0001$).

The online version of this article includes the following source data for figure 7:

**Source code 1.** Hepatocyte-specific MCT1KO accelerated fibrosis, while hepatic stellate cell-specific MCT1KO decreased it.

**Source data 1.** Hepatocyte-specific MCT1KO accelerated fibrosis, while hepatic stellate cell-specific MCT1KO decreased it.

(*Figure 7D and E*). Nonetheless, taken together, our data suggest that lactate flux via MCT1 in hepatic stellate cells strongly promotes collagen 1 translation or inhibits protein turnover rates (*Figure 8—figure supplement 3*).

Interestingly, the inhibitory effect of stellate cell MCT1KO was observed only at the collagen 1 protein level, while the hepatocyte MCT1KO affected both collagen mRNA and protein. The exact mechanism needs to be further investigated, but one possibility is that increased import of lactate into hepatic stellate cells serves as a precursor for glycine, proline, and hydroxyproline synthesis, as suggested in other systems such as cancer or pulmonary fibrosis (*Pérez-Escuredo et al., 2016*; *Hamanaka et al., 2019*). This could potentially explain why MCT1 depletion in stellate cells has a greater translational effect on collagen expression rather than transcriptional regulation.

Strong support for the above conclusions was obtained in an alternative experimental model of NASH—genetically obese ob/ob mice on a GAN diet (*Yenilmez et al., 2022*). In these experiments we employed chemically stabilized siRNA, taking advantage of RNA modifications 2-fluoro, 2-*O*-methyl ribose, and phosphorothioate backbone replacement to block siRNA degradation and immune responses as well as enhance in vivo delivery effectiveness and silencing longevity (*Behlke, 2006*; *Khvorova and Watts, 2017*). Conjugation of such siRNA compounds with GN promotes binding to the asialoglycoprotein receptor that is primarily expressed in hepatocytes at high levels, and we confirmed hepatocyte-selective gene silencing with such constructs (*Figure 3B–D*). Moreover, injection of GN-MCT1-siRNA did not affect MCT1 expression in other major tissues including inguinal white adipose tissue, gonadal white adipose tissue, brown adipose tissue, intestine, heart, lung, kidney, and spleen (*Figure 3—figure supplement 1G and H*). Hepatocyte-selective MCT1 silencing strongly upregulated collagen 1 in livers of ob/ob mice on a GAN diet (*Figure 5D–F*), as did hepatocyte-selective MCT1KO in the CDHFD mouse model (*Figure 7A*). In contrast, Chol-conjugated, chemically modified siRNA targeting MCT1 silenced the gene in all three liver cell types tested, including hepatic stellate cells (*Figure 3E–G*), and did attenuate liver collagen 1 expression (*Figure 5A and B*). This result is consistent with the idea that depleting MCT1 in stellate cells decreases collagen 1 production, as shown by MCT1KO in stellate cells (*Figure 7B*). The data in *Figure 3* and *Figure 3—figure supplement 1* also show the effectiveness of RNA interference (RNAi) in interrogating cell-specific processes in liver, and highlight the multiple advantages over traditional small molecule inhibitors in developing therapeutics (*Bumcrot et al., 2006*; *Aagaard and Rossi, 2007*). To date, five RNAi-based therapeutic agents have received FDA approvals targeting multiple disease areas, and a great many clinical trials of oligonucleotide therapeutics are in progress (*Padda et al., 2023*; *Traber and Yu, 2023*; *Zhu et al., 2022*; *Mullard, 2022*).

Analysis of gene expression profiles also showed that hepatic MCT1 positively regulates the levels of SREBP1 and ChREBP, major transcription factors regulating liver lipid metabolism, as well as their target DNL genes. These effects were apparently not sufficient to reverse severe steatosis in the genetically obese NASH mouse model (*Figure 4*), although a slight decrease in mean lipid droplet size was observed. The remaining steatosis may be attributed to the continuous supply of fatty acids from adipose tissue lipolysis, which accounts for up to 65% of hepatic fat accumulation as opposed to only 25% coming from hepatic DNL (*Donnelly et al., 2005*). While recognizing that the decreased expression of DNL genes does not necessarily indicate an inhibited fatty acid synthesis rate, we also cannot rule out the possibility of compensatory effects from other MCT isoforms that are expressed. However, since MCT1 haploinsufficiency showed greatly reduced HFD-induced hepatic steatosis, the discrepancies with our data may be due to other tissues being involved or the different mouse models used. Overall, it is clear from our studies that steatosis is not much affected by hepatic MCT1

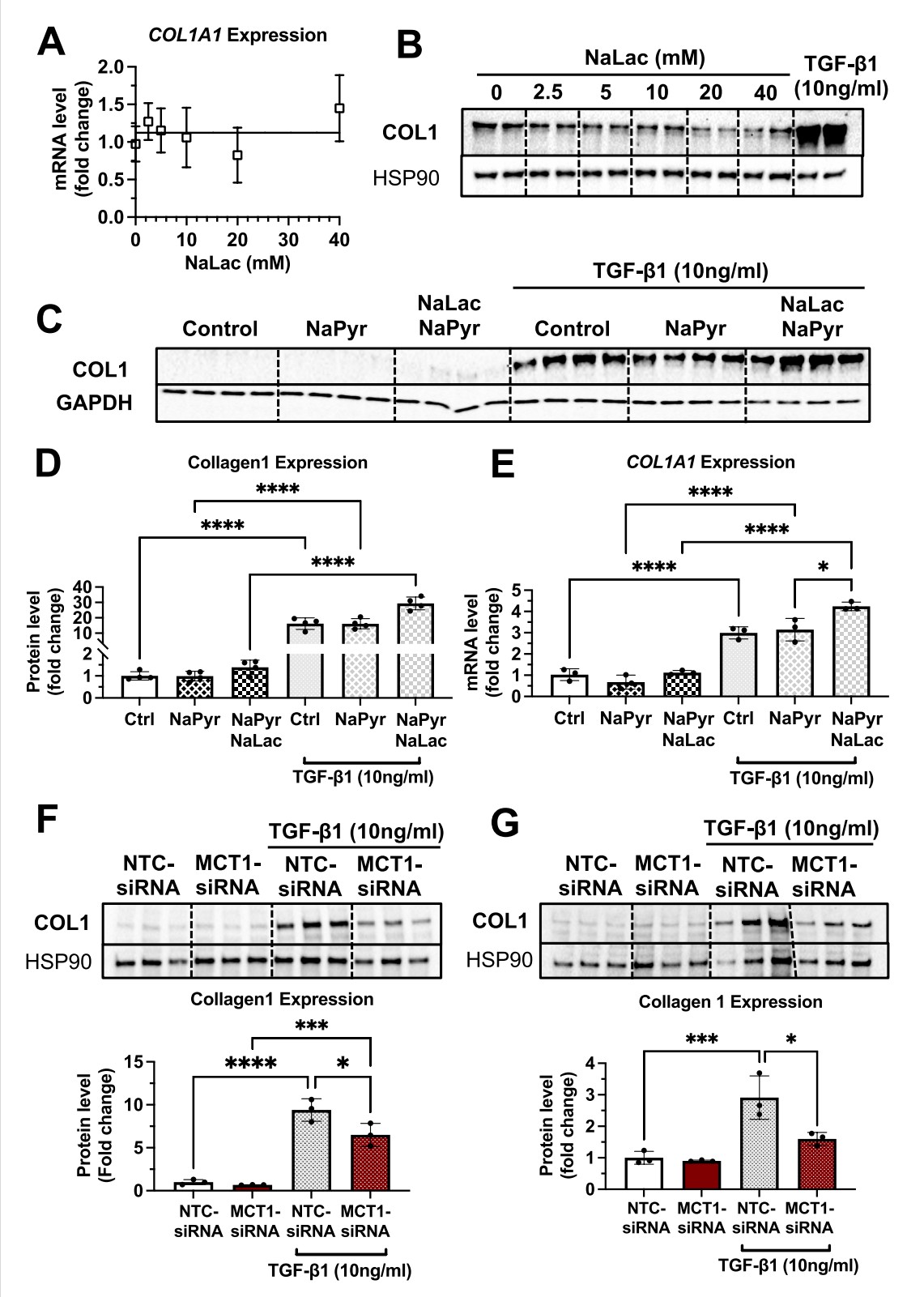

**Figure 8.** Lactate enhances the transforming growth factor 1β (TGF-β1)-stimulatory effect in the presence of pyruvate in human LX2 stellate cells. Cells were treated with increasing doses of sodium lactate (0, 2.5, 5, 10, 20, 40 mM) for 48 hr. Dose-response effect of sodium lactate on (**A**) *COL1A1* mRNA and (**B**) collagen 1 protein levels were monitored. Lactate effect on TGF-β1-stimulated collagen production was also examined. LX2 cells were treated with three conditions with or without TGF-β1 (10 ng/ml) treatment for 48 hr: (1) control (DMEM/high glucose media only), (2) sodium pyruvate (1 mM),

*Figure 8 continued on next page*

*Figure 8 continued*

and (3) the combination of sodium pyruvate (1 mM) and sodium lactate (10 mM). (**C**) Collagen 1 protein levels. (**D**) Quantification of collagen 1 protein levels. (**E**) COL1A1 mRNA expression levels. To test if MCT1 depletion can inhibit both endogenous and exogenous lactate-mediated collagen 1 production, cells were transfected with either NTC-siRNA or MCT1-siRNA for 6 hr. Then, cells were maintained in serum-starved media with or without 10 ng/ml of recombinant human TGF-β1 for 48 hr and harvested. (**F**) Collagen 1 protein levels upon sodium pyruvate-deprived condition. (**G**) Collagen 1 protein levels upon sodium pyruvate (1 mM) and sodium lactate (10 mM) condition (mean ± SD, one-way ANOVA, *: p<0.05, **: p<0.01, ***: p<0.001, ****: p<0.0001).

The online version of this article includes the following source data and figure supplement(s) for figure 8:

**Source data 1.** Lactate enhances the transforming growth factor 1β (TGF-β1)-stimulatory effect in the presence of pyruvate in human LX2 stellate cells.

**Figure supplement 1.** MCT1 promotes SMAD3 phosphorylation/activation in human LX2 stellate cells.

**Figure supplement 1—source data 1.** MCT1 promotes SMAD3 phosphorylation/activation in human LX2 stellate cells.

**Figure supplement 2.** MCT1 silencing enhanced fibrogenic gene expression levels in human hematoma cell lines, HepG2.

**Figure supplement 2—source data 1.** MCT1 silencing enhanced fibrogenic gene expression levels in human hematoma cell lines, HepG2.

**Figure supplement 3.** Graphical abstract.

depletion in two models of NASH in mice. A possible future strategy for NASH therapeutics may be to combine the depletion of MCT1 in stellate cells to decrease fibrosis with a potent anti-steatosis target such as DGAT2 (*Yenilmez et al., 2022*; *Calle et al., 2021*; *Loomba et al., 2020*) which we and others have shown is effective in reducing steatosis when depleted. Although there are still remaining challenges in developing a successful hepatic stellate cell-selective delivery system (*Poelstra, 2020*), ongoing studies are identifying promising candidates such as M6P-polyethylene glycol (*Zhu and Mahato, 2010*), IGF2R-specific peptide coupled nanocomplex (*Zhao et al., 2018*), and others (*Chen et al., 2019*).

To understand the underlying mechanism by which hepatic MCT1 depletion drives the downregulation of DNL genes expression, we investigated AMPK activation, as pAMPK has a negative regulatory effect on SREBP1 and ChREBP activation (*Li et al., 2011*; *Kawaguchi et al., 2002*). Indeed, hepatic MCT1 silencing enhanced AMPK phosphorylation, consistent with the previous MCT1 haploinsufficient mice study (*Carneiro et al., 2017*). These data are also in line with another study in which MCT1 inhibition reduced ATP production and activated AMPK, thus deactivating SREBP1c and lowering levels of its target SCD1 (*Zhao et al., 2020*). It remains to be examined whether other mechanisms are also at play that connect MCT1 function in liver to DNL gene regulation.

In summary, the data presented here highlight hepatic stellate cell MCT1 as a potential therapeutic target to prevent NASH fibrogenesis related to collagen 1 production. Its utility as a therapeutic target is complicated by our finding that MCT1 depletion in hepatocytes actually increases fibrosis. This work highlights the importance of contemplating cell-type specificity when developing therapeutic strategies, especially in systems of complex cellular landscapes such as NASH.

## Materials and methods
### Oligonucleotide synthesis

The 15–20 or 18–20 sequence oligonucleotides were synthesized by phosphoramidite solid-phase synthesis on a Dr Oligo 48 (Biolytic, Fremont, CA, USA), or MerMade12 (Biosearch Technologies, Novato, CA, USA) using 2′-F and 2′-O-Me phosphoramidites with standard protecting groups (Chemgenes, Wilmington, MA, USA). For the 5′-VP coupling, 5'-(*E*)-vinyl tetraphosphonate (pivaloyloxymethyl) 2'-*O*-methyl-uridine 3'-CE phosphoramidite was used (Hongene Biotech, Union City, CA, USA). Phosphoramidites were prepared at 0.1 M in anhydrous acetonitrile (ACN), except for 2'-*O*-methyl-uridine phosphoramidite dissolved in anhydrous ACN containing 15% dimethylformamide. 5-(Benzylthio)1*H*-tetrazole (BTT) was used as the activator at 0.25 M, coupling time for all phosphoramidites was 4 min. Detritylations were performed using 3% trichloroacetic acid in dichloromethane. Capping reagents used were CAP A (20% *n*-methylimidazole in ACN) and CAP B (20% acetic anhydride and 30% 2,6-lutidine in ACN). Phosphite oxidation to convert to phosphate or phosphorothioate was performed with 0.05 M iodine in pyridine-$H_2O$ (9:1, vol/vol) or 0.1 M solution of 3-[(dimethylaminomethylene)amino]-3*H*-1,2,4-dithiazole-5-thione (DDTT) in pyridine, respectively. All synthesis reagents were purchased from ChemGenes. Unconjugated oligonucleotides were

synthesized on 500 Å long-chain alkyl amine (LCAA) controlled pore glass (CPG) functionalized with Unylinker terminus (ChemGenes). Chol-conjugated oligonucleotides were synthesized on a 500 Å LCAA-CPG support, functionalized with a tetra-ethylenglycol cholesterol moiety bound through a succinate linker (ChemGenes). GN-conjugated oligonucleotides were grown on a 500 Å LCAA-CPG functionalized with an aminopropanediol-based trivalent GalNAc cluster (Hongene).

## Deprotection and purification of oligonucleotides for screening of sequences

Prior to the deprotection, synthesis columns containing oligonucleotides were treated with 10% diethylamine (DEA) in ACN to deprotect cyanoethyl groups. Synthesis columns containing the oligonucleotides covalently attached to the solid supports were cleaved and deprotected for 1 hr at room temperature with anhydrous mono-methylamine gas (Airgas). Columns with deprotected oligonucleotides were washed with 1 ml of 0.1 M sodium acetate in 85% ethanol aqueous solution, followed by rinse with an 85% ethanol aqueous solution. The excess ethanol was dried from the column on a vacuum manifold. Finally, the oligonucleotides were eluted off the columns with MilliQ water.

## Deprotection and purification of oligonucleotides for in vivo experiments

Chol- or GN-conjugated oligonucleotides were cleaved and deprotected with 28–30% ammonium hydroxide and 40% aqueous methylamine in a 1:1 ratio for 2 hr at room temperature. VP-containing oligonucleotides were cleaved and deprotected as described previously. Briefly, CPG with VP-oligonucleotides was treated with a solution of 3% DEA in 28–30% ammonium hydroxide for 20 hr at 35°C. The cleaved oligonucleotide solutions were filtered to remove CPG and dried under a vacuum. The pellets were resuspended in 5% ACN in water and purified on an Agilent 1290 Infinity II HPLC system. VP and GN-conjugated oligonucleotides were purified using a custom 20×150 mm$^2$ column packed with Source 15Q anion exchange resin (Cytiva, Marlborough, MA, USA). Run conditions were the following. Eluent A: 20 mM sodium acetate in 10% ACN in water. Eluent B: 1 M sodium bromide in 10% ACN in water. Linear gradient 10–35% B 20 min at 40°C. Chol-conjugated oligonucleotides were purified using 21.2×150 mm$^2$ PRP-C18 column (Hamilton Co, Reno, NV, USA). Run conditions were the following: Eluent A, 50 mM sodium acetate in 5% ACN in water; Eluent B: 100% ACN. Linear gradient, 40–60% B 20 min at 60°C. Flow used was 40 ml/min for both systems. Peaks were monitored at 260 nm. Fractions collected were analyzed by liquid chromatography-mass spectrometry (LC-MS). Pure fractions were combined and dried under a vacuum and resuspended in 5% ACN. Oligonucleotides were desalted by size exclusion on a 50×250 mm$^2$ custom column packed with Sephadex G-25 media (Cytiva, Marlborough, MA, USA), and lyophilized. Reagents for deprotection and purification were purchased from Fisher Scientific, Sigma-Aldrich, and Oakwood Chemicals.

## LC-MS analysis of oligonucleotides

The identity of oligonucleotides is verified by LC-MS analysis on an Agilent 6530 accurate mass Q-TOF using the following conditions: buffer A: 100 mM 1,1,1,3,3,3-hexafluoroisopropanol (HFIP) (Oakwood Chemicals) and 9 mM triethylamine (TEA) (Fisher Scientific) in LC-MS grade water (Fisher Scientific); buffer B:100 mM HFIP and 9 mM TEA in LC-MS grade methanol (Fisher Scientific); column, Agilent AdvanceBio oligonucleotides C18; linear gradient 0–35% B 5 min was used for unconjugated and GN-conjugated oligonucleotides; linear gradient 50–100% B 5 min was used for cholesterol-conjugated oligonucleotides; temperature, 60°C; flow rate, 0.85 ml/min. LC peaks are monitored at 260 nm. MS parameters: source, electrospray ionization; ion polarity, negative mode; range, 100–3,200 m/z; scan rate, 2 spectra/s; capillary voltage, 4000; fragmentor, 200 V; gas temperature, 325°C.

## LX2 human hepatic stellate cell studies

Human hepatic stellate cell line, LX2, was freshly purchased from the Millipore Sigma (cat SCC064) and the cell line authentication test conducted by ATCC demonstrated a match percentage exceeding the 80% threshold. LX2 cells were cultured in DMEM/high glucose media (Gibco, cat 11995065 and Fisher, cat 11965092) with 10% FBS. To test the preventative effect of MCT1 depletion in TGF-β1-stimulated hepatic stellate cell conditions, LX2 cells were plated in 6-well plates (300k cells/well) or 12-well plates (150k cells/well) in DMEM/high glucose media with 2% FBS. To test the lactate

effect on collagen production, cells were treated with sodium lactate (Sigma-Aldrich, cat L7022) for 48 hr and harvested. To test the MCT1 effect on TGF-β1-stimulated collagen production, cells were first transfected with either NTC-siRNA or MCT1-siRNA (IDT, cat 308915476) using Lipofectamine RNAi Max (Thermo Fisher, cat 13778075) for 6 hr in less serum optiMEM media (Thermo Fisher, cat 31985062). Then, cells were maintained in serum-starved media with or without 10 ng/ml of recombinant human TGF-β1 (R&D Systems, cat 240-B/CF) for 48 hr and harvested. As a housekeeping gene, β-actin (*ACTB*) was used.

## Human HepG2 hepatoma cell studies

Human hepatoma cell line, HepG2, was freshly purchased from ATCC (cat HB-8065) and the cell line authentication test conducted by ATCC confirmed a match percentage exceeding the 80% threshold. Cells were cultured in RPMI media (Gibco, cat 11875-093) with 10% FBS. To test the effect of MCT1 depletion, cells were plated in 6 well plates (300k cells/well) or 12-well plates (150k cells/well). The next day, cells were transfected with either NTC-siRNA or MCT1-siRNA using Lipofectamine RNAi Max (Thermo Fisher, cat 13778075) for 6 hr in less serum optiMEM media (Thermo Fisher, cat 31985062). After 48 hr, HepG2 cells were harvested. The media were saved to further test for secreted factors that may affect hepatic stellate cell activation. LX2 cells were incubated with the conditioned media (40% conditioned media+60% fresh media), and cells were harvested after 48 hr.

## In vitro screening of chemically modified siRNAs

Mouse hepatocyte cell line, FL83B, was freshly purchased from ATCC (cat CRL-2390) and the cell line authentication test conducted by ATCC confirmed a match percentage exceeding the 80% threshold. FL83B cells were plated in 12-well plates (150k cells/well) in F-12K medium with 3% FBS. Then, 1.5 μM of each Chol-MCT1-siRNA candidate compound was added and Chol-NTC-siRNA was used as a control. Then, 72 hr after the treatment, cells were harvested, and the *Mct1* mRNA silencing potency was monitored. To further evaluate the half maximal inhibitory concentration (IC50) values, the dose-dependent silencing effect of the compounds was calculated upon six different concentrations (1.5, 0.75, 0.38, 0.19, 0.05, and 0 μM). As a housekeeping gene, β-2-microglobulin (*B2m*) was used.

## Generation and validation of hepatic stellate cell-specific AAV9-Lrat-Cre

Lrat-Cre-mediated KO mice have been widely utilized in the field to delete genes in hepatic stellate cells (*Mederacke et al., 2013*). We newly synthesized AAV9-Lrat-Cre to establish an inducible hepatic stellate cell KO system in collaboration with Vector Biolabs. Proximal mouse Lrat promoter region from −1166 bp, including the putative transcriptional start site, to +262 bp downstream sequence was chosen (*Prukova et al., 2015*). A 1428b Lrat promoter was synthesized and cloned into Vector Biolabs' AAV-CMV-Cre vector to replace CMV promoter with Lrat promoter. The AAV-Lrat-Cre was then packaged into AAV9 virus. As a control, AAV-Lrat-null constructs were used.

## Animal studies

All animal procedures were performed in accordance with animal care ethics approval and guidelines of University of Massachusetts Chan Medical School Institutional Animal Care and Use Committee (IACUC, protocol number A-1600-19). All wild-type C57BL6/J male mice and genetically obese ob/ob male mice were obtained from Jackson Laboratory. MCT1^fl/fl mice were generated in the Rothstein lab (*Jha et al., 2020*). Mice were group-housed on a 12 hr light/dark cycle and had ad libitum access to water and food. For each experiment, mice are randomly assigned to control and experimental groups to ensure unbiased results. For obese NASH model studies, 10-week-old genetically obese ob/ob male mice (n=6) were subcutaneously injected with 10 mg/kg of siRNAs accordingly (Chol-NTC-siRNA, Chol-MCT1-siRNA, GN-NTC-siRNA, and GN-MCT1-siRNA), every 10–12 days. Mice were fed the GAN diet (Research Diets, cat D09100310) for 3 weeks. Food intake and body weight were monitored. Mice were sacrificed with $CO_2$, and double-killed with cervical dislocation. For CDAHFD-induced NASH model studies, 8-week-old male MCT1^fl/fl mice (n=10) were intravenously injected with $2×10^{11}$ gc of AAV-TBG-Cre or AAV-Lrat-Cre or both. As a control, the same amount of AAV-TBG-null or AAV-Lrat-null control was used. A week after the injection, mice were fed a CDHFD (Research Diets, cat A06071302i) for 8 weeks and sacrificed.

## Primary mouse cell isolation

Male C57BL/6 wild-type mice 16- to 18-week-old (n=4) were subcutaneously injected with 10 mg/kg of siRNAs accordingly (Chol-NTC-siRNA, Chol-MCT1-siRNA, GN-NTC-siRNA, and GN-MCT1-siRNA), twice within 15 days. Mice were put on a chow diet (LabDiet, cat 5P76) and sacrificed on day 15. Primary hepatocytes, hepatic stellate cells, and Kupffer cells were isolated from the livers using the modified perfusion method described previously (*Mederacke et al., 2015*; *Aparicio-Vergara et al., 2017*). Briefly, livers were digested in situ with 14 mg pronase (Sigma-Aldrich, cat P5147) and 3.7 U collagenase D (Roche, cat 11 088 882 001) via inferior vena cava. Digested livers were isolated and minced with 0.5 mg/ml pronase, 0.088 U/ml collagenase, and 0.02 mg/ml DNase I (Roche, cat 10 104 159 001). After centrifuging cells for 3 min at 50×*g* at 4°C, primary hepatocytes were obtained in the pellet. The remaining supernatant was collected and centrifuged for 10 min at 580×*g* at 4°C and the pellet was saved for further hepatic stellate cell separation using Nycodenz (Accurate Chemical, cat 1002424) gradient solution. Lastly, Kupffer cells were isolated from the remaining cells using a Percoll (Sigma, cat P1644) gradient solution. Separation of hepatocytes, hepatic stellate cells, and Kupffer cells was validated using representative mRNA markers of each cell type such as *Alb, Des*, and *Clec4f*, respectively, by rt-qPCR.

## Serum analysis

Retro-orbital bleeding was performed prior to sacrificing mice. Blood was collected in heparinized capillary tubes and centrifuged 10 min at 7000 rpm at 4°C. Supernatant plasma was saved for further serum analysis. Plasma lactate level was measured using a specific apparatus, Lactate Plus meter (Nova Biomedical, cat 62624). ALT level was determined using ALT Colorimetric Activity Assay Kit (Cayman, cat 700260). Absorbances were detected using a Tecan safire2 microplate reader.

## Glucose tolerance test

GTT was performed after 16 hr of fasting. Basal glucose level was measured using a glucometer (CONTOUR NEXT ONE glucose meter), then mice were intraperitoneally injected with 1 g/kg body weight D-glucose dissolved in sterile saline. Blood glucose was measured with a single drop of tail blood at 15, 30, 45, 90, and 120 min after the glucose injection.

## Shear wave elastography

Mouse liver stiffness was monitored by Vega robotic ultrasound imager, SonoEQ 1.14.0 (SonoVol), as described in a previous study (*Morin et al., 2021*). Before SWE measurement, mice had their abdomen hair shaved and the residual hair was removed using chemical depilation cream (Nair). After being anesthetized with isoflurane, mice were located in prone position on the fluid chamber through an acoustically transmissive membrane with ultrasound transducer imaging from below. During the imaging, wide-field B-mode was captured, a 3D volume was reconstructed, liver was visualized, and fiducial markers in 3D space indicating the position of the desired SWE capture were placed. Liver stiffness was monitored by Young's modulus.

## RNA isolation and rt-qPCR

Frozen mouse livers samples (25 mg) or in vitro cell samples were homogenized in Trizol (Ambion) using QIAGEN TissueLyser II. Chloroform was added and centrifuged for 15 min at maximum speed at 4°C. The supernatant was collected and 100% isopropanol was added. After another 10 min centrifugation at maximum speed at 4°C, the pellet was saved and washed with 70% ethanol with 5 min centrifugation at maximum speed at 4°C. The pellet was dried briefly and resuspended with ultrapure distilled water (Invitrogen). cDNA was synthesized using 1 µg of total RNA using iScript cDNA Synthesis Kit (Bio-Rad) on Bio-Rad T100 thermocycler. rt-qPCR was performed using iQ SybrGreen Supermix on CFX96 1000 thermocycler (Bio-Rad) and analyzed as described (*Livak and Schmittgen, 2001*). Primer sequences used for rt-qPCR were listed in *Table 3*.

## Immunoblotting

Frozen mouse liver samples (50 mg) or in vitro cell samples were homogenized in a sucrose lysis buffer (250 mM sucrose, 50 mM Tris-Cl pH 7.4) with 1:100 phosphatase and protease inhibitor cocktail (Sigma-Aldrich) using QIAGEN TissueLyser II. Protein concentration was determined by BCA assay.

**Table 3.** List of primers used for real-time quantitative PCR (rt-qPCR).

**Mouse primers:**

| Gene | Forward | Reverse |
| --- | --- | --- |
| Slc16a1/Mct1 | TGTTAGTCGGAGCCTTCATTTC | CACTGGTCGTTGCACTGAATA |
| Slc16a1Mct1 (Exon 2,3 overlapping) | TGCAACGACCAGTGAAGTATC | GCTGCCGTATTTATTCACCAAG |
| Slc16a7/Mct2 | CCATCAGTAGTGTGTTGGTGAA | TCTATCACGCTGTTGCTGTAAG |
| Slc16a3/Mct4 | AGTGCCATTGGTCTCGTG | CATACTTGTAAACTTTGGTTGCATC |
| Srebf1 | GGAGCCATGGATTGCACATT | GGCCCGGGAAGTCACTGT |
| Mlxipl | TCTGCAGATCGCGTGGAG | CTTGTCCCGGCATAGCAAC |
| Fasn | GGAGGTGGTGATAGCCGGTAT | TGGGTAATCCATAGAGCCCAG |
| Scd1 | CCGGAGACCCCTTAGATCGA | TAGCCTGTAAAAGATTTCTGCAAACC |
| Acly | TGGTGGAATGCTGGACAA | GCCCTCATAGACACCATCTG |
| Tgfb1 | CTCCCGTGGCTTCTAGTGC | GCCTTAGTTTGGACAGGATCTG |
| Ihh | CTCTTGCCTACAAGCAGTTCA | CCGTGTTCTCCTCGTCCTT |
| Acta2 | ATGCTCCCAGGGCTGTTTTCC | GTGGTGCCAGATCTTTTCCATGTCG |
| Gli2 | CAACGCCTACTCTCCCAGAC | GAGCCTTGATGTACTGTACCAC |
| Gli3 | CACAGCTCTACGGCGACTG | CTGCATAGTGATTGCGTTTCTTC |
| Col1a1 | GCTCCTCTTAGGGGCCACT | CCACGTCTCACCATTGGGG |
| Col1a2 | GTAACTTCGTGCCTAGCAACA | CCTTTGTCAGAATACTGAGCAGC |
| Col3a1 | CTGTAACATGGAAACTGGGGAAA | CCATAGCTGAACTGAAAACCACC |
| Timp1 | CTCAAAGACCTATAGTGCTGGC | CAAAGTGACGGCTCTGGTAG |
| Alb | TGCTTTTTCCAGGGGTGTGTT | TTACTTCCTGCACTAATTTGGCA |
| Des | CTAAAGGATGAGATGGCCCG | GAAGGTCTGGATAGGAAGGTTG |
| Clec4f | GAGGCCGAGCTGAACAGAG | TGTGAAGCCACCACAAAAAGAG |
| B2m | CATGGCTCGCTCGGTGAC | CAGTTCAGTATGTTCGGCTTCC |
| F4/80 | CTTTGGCTATGGGCTTCCAGTC | GCAAGGAGGACAGAGTTTATCGTG |
| Ccl2 | AGGTCCCTGTCATGCTTCTG | AAGGCATCACAGTCCGAGTC |
| Il1b | TTTGACAGTGATGAGAATGACC | CTCTTGTTGATGTGCTGCTG |
| Tlr4 | ATGGCATGGCTTACACCACC | GAGGCCAATTTTGTCTCCACA |
| Ccr2 | ATCCACGGCATACTATCAACATC | CAAGGCTCACCATCATCGTAG |
| Ccl20 | GCCTCTCGTACATACAGACGC | CCAGTTCTGCTTTGGATCAGC |
| Cd163 | ATGGGTGGACACAGAATGGTT | CAGGAGCGTTAGTGACAGCAG |
| Arg1 | CTCCAAGCCAAAGTCCTTAGAG | AGGAGCTGTCATTAGGGACATC |
| Ccl22 | AGGTCCCTATGGTGCCAATGT | CGGCAGGATTTTGAGGTCCA |
| Cd206 | CTCTGTTCAGCTATTGGACGC | TGGCACTCCCAAACATAATTTGA |

**Human primers:**

| Gene | Forward | Reverse |
| --- | --- | --- |
| SLC16A1/MCT1 | TGGAAGACACCCTAAACAAGAG | AAAGCCTCTGTGGGTGAATAG |
| ACTA2 | AGCGTGGCTATTCCTTCGT | CTCATTTTCAAAGTCCAGAGCTACA |
| TGFB1 | CAACGAAATCTATGACAAGTTCAAGCAG | CTTCTCGGAGCTCTGATGTG |
| COL1A1 | ACGTCCTGGTGAAGTTGGTC | ACCAGGGAAGCCTCTCTCTC |

*Table 3 continued on next page*

Table 3 continued

**Human primers:**

| TIMP1 | AATTCCGACCTCGTCATCAGG | ATCCCCTAAGGCTTGGAACC |
|---|---|---|
| ACTB | GATGAGATTGGCATGGCTTT | GAGAAGTGGGGTGGCTT |

Immunoblotting loading samples were prepared after adjusting the protein concentration using 5× SDS (Sigma-Aldrich) and denatured by boiling. Proteins were separated in 4–15% sodium dodecyl sulfate/polyacrylamide gel electrophoresis gel (Bio-Rad) and transferred to nitrocellulose membranes. The membranes were blocked with Tris-buffered saline with Tween (TBST) with 5% skim milk or 5% bovine serum albumin. Membranes were incubated with primary antibodies overnight at 4°C, washed in TBST for 30 min, then incubated with secondary antibodies for an hour at room temperature, and washed for 30 min in TBST. Antibodies used in the studies were listed in *Table 4*. ECL (Perkin Elmer) was added to the membranes and the protein signals were visualized with ChemiDox XRS+image-forming system.

## Immunofluorescence
FL83B cells were seeded in coverslips placed in 12-well plates (150k cells/well). Then, 1.5 µM of either Chol-NTC-siRNA or Chol-MCT1-siRNA at final concentration was added for 72 hr. To stain mitochondrial membranes, cells were incubated with Mitotracker at 37°C (Thermo Fisher, cat M7512) for 45 min in serum-free media. Then, cells were fixed with 4% paraformaldehyde at room temperature

**Table 4.** List of antibodies used in this study.

| Reagent | Source | Identifier |
|---|---|---|
| Anti-MCT1 | Proteintech | Cat # 20139-1-AP |
| Anti-FASN | Cell Signaling | Cat # 3180s |
| Anti-ACLY | Cell Signaling | Cat # 4332 |
| Anti-SCD1 | Cell Signaling | Cat # 2794s |
| Anti-ChREBP | Novus Bio | Cat # NB400-135 |
| Anti-SREBP1 | Millipore | Cat # MABS1987 |
| Anti-GAPDH-HRP | Cell Signaling | Cat # 8884s |
| Anti-H3 | Cell Signaling | Cat # 4499s |
| Anti-Tubulin | Sigma-Aldrich | Cat # T5168 |
| Anti-pAMPK (T172) | Cell Signaling | Cat # 2535s |
| Anti-AMPKα | Cell Signaling | Cat # 2793s |
| Anti-αSMA | Cell Signaling | Cat # 19,245s |
| Anti-Collagen 1 | Southern Biotech | Cat # 1310–01 |
| Anti-HSP90-HRP | Cell Signaling | Cat # 79,631s |
| Anti-SMAD3 | Cell Signaling | Cat # 9523S |
| Anti-pSMAD3 | Abcam | Cat # AB52903 |
| Goat Anti-Rabbit IgG-HRP | Invitrogen | Cat # 65-6120 |
| Goat Anti-Mouse IgG-HRP | Invitrogen | Cat # 65-6520 |
| Goat Anti-Mouse IgG-HRP | Thermo Fisher | Cat # G21040 |
| Mouse Anti-Goat IgG-HRP | Santa Cruz | Cat # sc-2354 |
| Goat-anti-Rabbit-488 | Thermo Fisher | Cat # A11008 |
| ProLong Gold Antifade Mountant | Thermo Fisher | Cat # P36931 |

for 30 min. Fixed cells were blocked by fresh permeabilization buffer (0.5% Triton, 1% FBS in PBS) at room temperature for 30 min and incubated with 1:100 Anti-MCT1 (Proteintech, cat 20139-1-AP) overnight at 4°C. As a secondary antibody, 1:1000 goat-anti-Rabbit-488 was used, while cells were protected from light. Coverslips were mounted on ProLong Gold Antifade Mountant with DAPI (Invitrogen, cat P35934). Images were acquired using an Olympus IX81 microscope (Central Valley, PA, USA) with dual Andor Zyla sCMOS 4.2 cameras (Belfast, UK). Images were quantified using ImageJ software.

### Histological analysis

For the immunohistochemistry, half of the biggest lobe of each mouse liver was fixed in 4% paraformaldehyde and embedded in paraffin. Sectioned slides were stained where indicated with H&E, Trichrome, Sirius Red, and anti-MCT1 (Proteintech, cat 20139-1-AP) at the University of Massachusetts Chan Medical School Morphology Core. The whole stained slides were scanned with ZEISS Axio Scan Z1. Images were analyzed by ZEN 3.0 and ImageJ software.

### H&E lipid droplet analysis

To quantify the mean size and the mean number of lipid droplets, H&E images further underwent thorough image analysis. The 2D RGB images (8 bits per channel) were read into the Fiji version (*Schindelin et al., 2012*) based on ImageJ2 (*Rueden et al., 2017*). An ImageJ macro language program was then written to analyze each image. First, the Labkit plugin (*Arzt et al., 2022*) was used to classify pixels as either lipid or background. The classifier used was trained on a few short line segments drawn in either lipid or background regions. The binary objects created then had their holes filled and were culled using 'Analyze Particles' to eliminate objects larger than 10,000 pixels (typically veins). Then the 'watershed' algorithm was used to separate touching lipid droplets, and 'Analyze Particles' was used again, this time to keep objects with a circularity of 0.5–1 and a size of 40–5000 pixels. Pixel size was converted to µm using a 0.47 µm/pixel width conversion ratio (so a 0.22 µm$^2$/pixel conversion factor).

### Trichrome and Sirius Red image analysis for fibrosis

To quantify the % of fibrotic regions, 2D RGB images (8 bits per channel) of Sirius Red and Trichrome (without the hematoxylin stain) were read into the Fiji version (*Schindelin et al., 2012*) based on ImageJ2 (*Rueden et al., 2017*). 'Analyze Particles' was used to threshold the Sirius Red images (×20 magnification) in the green channel, keeping pixels with an intensity <100 and object size >100 pixels as fibrotic regions. Pixels in the Trichrome images (×2.5 magnification) in the red channel with intensity <60 and object size >0.0005 pixels were considered as fibrotic regions.

### Quantification and statistical analysis

All statistical analyses were calculated using GraphPad Prism 9 (GraphPad software). A two-sided unpaired Student's t-test was used for the analysis of the statistical significance between the two groups. For more than three groups, one-way ANOVA was used for the analysis of statistical significance. Data were presented as mean ± SD or otherwise noted. Differences were considered significant when p<0.05 (*: p<0.05, **: p<0.005, and ***: p<0.0005). Data were excluded only when a technical error occurred in sample preparation or after identifying outliers through analysis in GraphPad Prism 9. Sample sizes were decided based on previous publications or power analysis. Statistical significance was pursued through the utilization of a minimum of three technical replicates and three biological replicates, with each graph incorporating the representation of individual data points.

## Acknowledgements

We thank all members of Michael Czech's Lab and Dr. Zinger Yang Loureiro for helpful discussions and critical reading of the manuscript. We thank the UMASS Morphology Core for assistance in immunohistochemistry analysis. The graphical abstract was created with BioRender.com. This work was supported by National Institutes of Health grants DK116056, DK103047, and DK030898 to MPC.

## Additional information

### Competing interests

Michael P Czech: Reviewing editor, eLife. The other authors declare that no competing interests exist.

### Funding

| Funder | Grant reference number | Author |
|---|---|---|
| National Institutes of Health | DK116056 | Kyounghee Min<br>Batuhan Yenilmez<br>Mark Kelly<br>Michael Elleby<br>Lawrence M Lifshitz<br>Naideline Raymond<br>Emmanouela Tsagkaraki<br>Shauna M Harney<br>Chloe DiMarzio<br>Hui Wang<br>Michael P Czech |
| National Institutes of Health | DK103047 | Kyounghee Min<br>Batuhan Yenilmez<br>Mark Kelly<br>Michael Elleby<br>Lawrence M Lifshitz<br>Naideline Raymond<br>Emmanouela Tsagkaraki<br>Shauna M Harney<br>Chloe DiMarzio<br>Hui Wang<br>Michael P Czech |
| National Institutes of Health | DK030898 | Kyounghee Min<br>Batuhan Yenilmez<br>Mark Kelly<br>Michael Elleby<br>Lawrence M Lifshitz<br>Naideline Raymond<br>Emmanouela Tsagkaraki<br>Shauna M Harney<br>Chloe DiMarzio<br>Hui Wang<br>Michael P Czech |

The funders had no role in study design, data collection and interpretation, or the decision to submit the work for publication.

### Author contributions

Kyounghee Min, Conceptualization, Resources, Data curation, Software, Formal analysis, Validation, Investigation, Visualization, Methodology, Writing – original draft, Project administration, Writing – review and editing; Batuhan Yenilmez, Conceptualization, Resources, Formal analysis, Investigation; Mark Kelly, Resources, Investigation, Methodology; Dimas Echeverria, Resources, Software, Validation; Michael Elleby, Shauna M Harney, Formal analysis; Lawrence M Lifshitz, Data curation, Software, Formal analysis, Validation, Visualization; Naideline Raymond, Resources, Formal analysis; Emmanouela Tsagkaraki, Chloe DiMarzio, Hui Wang, Formal analysis, Investigation; Nicholas McHugh, Brianna Bramato, Resources, Validation; Brett Morrison, Jeffery D Rothstein, Resources; Anastasia Khvorova, Resources, Software, Supervision, Validation; Michael P Czech, Conceptualization, Resources, Supervision, Funding acquisition, Investigation, Methodology, Writing – original draft, Project administration, Writing – review and editing

### Author ORCIDs

Kyounghee Min ● http://orcid.org/0000-0002-2546-0236
Batuhan Yenilmez ● http://orcid.org/0000-0001-8798-3676
Anastasia Khvorova ● http://orcid.org/0000-0001-6928-8071
Michael P Czech ● https://orcid.org/0000-0003-4075-7350

### Ethics

All animal procedures were performed in accordance with animal care ethics approval and guidelines of University of Massachusetts Chan Medical School Institutional Animal Care and Use Committee (IACUC, protocol number A-1600-19).

Reviewer #1 (Public Review): https://doi.org/10.7554/eLife.89136.3.sa1
Reviewer #2 (Public Review): https://doi.org/10.7554/eLife.89136.3.sa2
Reviewer #3 (Public Review): https://doi.org/10.7554/eLife.89136.3.sa3
Author Response https://doi.org/10.7554/eLife.89136.3.sa4

## Additional files

### Supplementary files

• MDAR checklist

### Data availability

All data quantified during this study are included in the manuscript and the Materials and methods.

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
