## [Editor Report · eLife assessment]

This **convincing** manuscript represents a **valuable** advance in understanding the role of MCT1 – a transporter for lactate and other organic anions – in hepatocytes and hepatic stellate cells in the liver. The authors also generate exciting new tools to investigate hepatic stellate cell biology, and these may have much broader applications, but future studies are required to validate these new tools.

---

## [Referee Report · Reviewer #1 (Public Review)]

The authors put forth the hypothesis that hepatocyte and/or non-parenchymal liver MCT1 may be responsible for physiologic effects (lower body weight gain and less hepatic steatosis) in MCT1 global heterozygote mice. They generate multiple tools to test this hypothesis, which they combine with mouse diets that induce fatty liver, steatohepatitis and fibrosis. Novel findings include that deletion of hepatocyte MCT1 does not change liver lipid content, but increases liver fibrosis. Deletion of hepatic stellate cell (HSC) MCT1 does not substantially affect any liver parameter, but concomitant HSC MCT1 deletion does reverse fibrosis seen with hepatocyte MCT1 knockout or knockdown. In both models, plasma lactate levels do not change, suggesting that liver MCT1 does not substantially affect systemic lactate. In general, the data match conclusions of the manuscript, and the studies are well-conducted and well-described. Further work would be necessary to dissect mechanism of fibrosis with hepatocyte MCT1, and whether this is due to changes in local lactate (as speculated by the authors) or another MCT1 substrate. This would be important to understand this novel potential cross-talk between hepatocytes and HSCs.

A parallel and perhaps more important advance is the generation of new methodology to target HSC in mice, using modified siRNA and by transduction of AAV9-Lrat-Cre. Both methods would reduce the need to cross floxed mice with the Lrat-Cre allele, saving time and resources. These tools were validated to an extent by the authors, but not sufficiently to ensure that there is no cross-reactivity with other liver cell types. For example, AAV9-Lrat-Cre-transduced MCT1 floxed mice show compelling HSC but not hepatocyte Mct1 knockdown, but other liver cell types should be assessed to ensure specificity. This is particularly important as overall liver Mct1 decreased by ~30% in AAV9-Lrat-Cre-transduced mice, which may exceed HSC content of these mice, especially when considering a 60-70% knockdown efficiency. This same issue also affects Chol-MCT1-siRNA, which the authors demonstrate to affect hepatocytes and HSC, but likely affects other cell types not tested. As this is a new and potentially valuable tool, it would be important to assess Mct1 expression across more non-parenchymal cells (i.e. endothelial, cholangiocytes, immune cells) to determine penetration and efficacy.

---

## [Referee Report · Reviewer #2 (Public Review)]

In this study, the authors seek to answer two main questions: (1) Whether interfering with lactate availability in hepatocytes through depletion of hepatocyte specific MCT-1 depletion would reduce steatosis, and (2) Whether MCT-1 in stellate cells promote fibrogenesis. While the first question is based on the observation that haploinsufficiency of MCT-1 makes mice resistant to steatosis, the rationale behind how MCT-1 could impact fibrogenesis in stellate cells is not clear. A more detailed discussion regarding how lactate availability would regulate two different processes in two different cell types would be helpful. The authors employ several mouse models and in vitro systems to show that MCT1 inhibition in hepatic stellate cells reduces the expression of COL-1.

The authors have sufficiently addressed prior comments and added new experiments to provide details on possible mechanisms.

---

## [Referee Report · Reviewer #3 (Public Review)]

We commend the authors on addressing our points and do believe that the manuscript is much improved. Even with the added in vitro data (Figure 8, Supplementary Figure 6), however, a clear mechanistic explanation for how MCT1 is modulating/inhibiting fibrosis in hepatic stellate cells is lacking and this represents a key area for future exploration. The authors provide interesting follow up experiments that suggest lactate can potentiate TGF-β1 signaling, a phenomenon that has previously been described in pulmonary fibrosis. Additionally, MCT1 depletion decreased the pSMAD3/SMAD ratio, but this was overcome with higher doses of TGB- β1 ligand. It remains unclear how intracellular versus extracellular lactate is signaling to exert the observed effects, and how the altered metabolism/metabolic flux in NAFLD is contributing to organ level metabolic dysregulation. These will be keys questions to answer going forward potentially using the novel in vivo models that the authors have contributed here.

A major finding of this work is that loss of monocarboxylate transporter 1 (MCT1), specifically in stellate cells, can decrease fibrosis in the liver. However, the underlying mechanism whereby MCT1 influences stellate cells is not addressed. It is unclear if upstream/downstream metabolic flux within different cell types leads to fibrotic outcomes. Ultimately, the paper opens more questions than it answers: why does decreasing MCT1 expression in hepatocytes exacerbate disease, while silencing MCT1 in fibroblasts seems to alleviate collagen deposition? Mechanistic studies in isolated hepatocytes and stellate cells could enhance the work further to show the disparate pathways that mediate these opposing effects. The work highlights the complexity of cellular behavior and metabolism within a disease environment but does little to mechanistically explain it.

---

## [Author Response]

The following is the authors’ response to the original reviews.

**eLife assessment**
This important study extends insights on NAFLD and NASH regarding the role of plasma lactate levels using mice haplo-insufficient for the gene encoding lactate transporter MCT-1. While the evidence is largely convincing and the work significantly advances our understanding of the roles of distinct hepatic cell types in steatosis, a number of issues require attention and would best be solved by further experimentation.

RESPONSE: We agree with this assessment by eLife, and appreciate the reviewers’ view that the study is important and extends insights into liver disease.

**Public Reviews:**

**Reviewer #1 (Public Review):**
The authors put forth the hypothesis that hepatocyte and/or non-parenchymal liver MCT1 may be responsible for physiologic effects (lower body weight gain and less hepatic steatosis) in MCT1 global heterozygote mice. They generate multiple tools to test this hypothesis, which they combine with mouse diets that induce fatty liver, steatohepatitis and fibrosis. Novel findings include that deletion of hepatocyte MCT1 does not change liver lipid content, but increases liver fibrosis. Deletion of hepatic stellate cell (HSC) MCT1 does not substantially affect any liver parameter, but concomitant HSC MCT1 deletion does reverse fibrosis seen with hepatocyte MCT1 knockout or knockdown. In both models, plasma lactate levels do not change, suggesting that liver MCT1 does not substantially affect systemic lactate. In general, the data match the conclusions of the manuscript, and the studies are well-conducted and well-described. Further work would be necessary to dissect mechanism of fibrosis with hepatocyte MCT1, and whether this is due to changes in local lactate (as speculated by the authors) or another MCT1 substrate. This would be important to understand this novel potential cross-talk between hepatocytes and HSCs.A parallel and perhaps more important advance is the generation of new methodology to target HSC in mice, using modified siRNA and by transduction of AAV9-Lrat-Cre. Both methods would reduce the need to cross floxed mice with the Lrat-Cre allele, saving time and resources. These tools were validated to an extent by the authors, but not sufficiently to ensure that there is no cross-reactivity with other liver cell types. For example, AAV9-LratCre-transduced MCT1 floxed mice show compelling HSC but not hepatocyte Mct1 knockdown, but other liver cell types should be assessed to ensure specificity. This is particularly important as overall liver Mct1 decreased by ~30% in AAV9-Lrat-Cre-transduced mice, which may exceed HSC content of these mice, especially when considering a 60-70% knockdown efficiency. This same issue also affects Chol-MCT1-siRNA, which the authors demonstrate to affect hepatocytes and HSC, but likely affects other cell types not tested. As this is a new and potentially valuable tool, it would be important to assess Mct1 expression across more non-parenchymal cells (i.e. endothelial, cholangiocytes, immune cells) to determine penetration and efficacy.

RESPONSE: We appreciate the reviewer’s view that the new methods we describe represent an important advance. To ensure the specificity of our novel AAV-Lrat-Cre construct, it would be fair to test its distribution among all possible hepatic cell types, including endothelial cells, cholangiocytes, and other immune cells, as suggested. Our efforts in this study were primarily focused on the major cell types thought to contribute to NASH, namely hepatocytes, Kupffer cells, and in particular hepatic stellate cells. The reasons for this focus were:

1. Our primary goal was to investigate the role of MCT1 in hepatic fibrogenesis. According to Manderacke et al. (2013, Nature Comm), hepatic stellate cells account for the dominant proportion (82-96%) of myofibroblast progenitors, which produce collagen fibers. While there may be interesting roles of MCT1 in those other cell types, to elucidate MCT1's role in fibrogenesis, focusing on the dominant fibrogenic cell type, hepatic stellate cells, was the most appropriate approach for this goal.

2. Considering the proportion of each hepatic cell type in the liver, hepatocytes constitute the majority (60-70%), followed by endothelial cells (15%), immune cells (10%), and stellate cells (5%), among others.

3. The AAV-Cre system is highly specific to its promoter, in this case, Lrat, which has been well established in multiple previous studies to exhibit high specificity for hepatic stellate cells in the liver.We will certainly conduct more comprehensive biodistribution studies in the future, as we believe that our AAV-Lrat-Cre system could be a valuable tool in this field.

**Reviewer #2 (Public Review):**
In this study, the authors seek to answer two main questions: (1) Whether interfering with lactate availability in hepatocytes through depletion of hepatocyte specific MCT-1 depletion would reduce steatosis, and (2) Whether MCT-1 in stellate cells promote fibrogenesis. While the first question is based on the observation that haploinsufficiency of MCT-1 makes mice resistant to steatosis, the rationale behind how MCT-1 could impact fibrogenesis in stellate cells is not clear. A more detailed discussion regarding how lactate availability would regulate two different processes in two different cell types would be helpful. The authors employ several mouse models and in vitro systems to show that MCT1 inhibition in hepatic stellate cells reduces the expression of COL-1. The significance of the findings is moderately impacted due to the following considerations:

RESPONSE: We have included additional in vitro data in order to provide a more comprehensive discussion of MCT1's potential role in regulating collagen production. Please refer to the new Figure 8, Supplementary Figure 6, and the results section (Potential Mechanism). Also note that our original hypothesis was that depleting MCT1 specifically in hepatocytes would protect mice with MCT1 haploinsufficiency from liver lactate overload and NAFLD. Furthermore, we postulated that this protection might prevent NASH progression since lipotoxicity-driven hepatocyte damage is a central factor in NASH pathogenesis. However, our findings did not support this hypothesis.We found only one brief article (2015, Z Gastroenterol et al., "Functional effects of monocarboxylate transporter 1 expression in activated hepatic stellate cells") that discussed the potential role of MCT1 depletion in hepatic stellate cells in regulating collagen production or fibrosis, as mentioned in their abstract. Unfortunately, the DOI for this article is not functional, and the data cannot be located. Moreover, when we attempted to replicate their results, we were unable to do so, leading us to report our own findings in the current paper.

a. Fibrosis in human NAFLD is a significant problem as a predictor of liver related mortality and is associated with type 1 and type 3 collagen. However, the reduction in COL1 in stellate cells did not amount to a reduction in liver fibrosis even in cell specific KO (in Fig 7E, there is no indication of whether Sirius red staining was different between HSC KO and control mice- the authors mention a downward trend in the text). The authors postulate that type 1 COL may not be the more predominant form of fibrosis in the model. This does not seem likely, since the same ob/ob mouse model was used to determine that fibrosis was enhanced with hepatocyte specific MCT-1 KO and decreased with Chol MCT-1KO. Measurements of different types of collagens in their model and the effect of MCT-1 on different types could be more informative. In particular, although collagens are the structural building blocks for hepatic fibrosis, fibrosis can also be controlled by matrix remodeling factors such as Timp1, Serpine 1, PAI-1 and Lox.

RESPONSE: We monitored the expression levels of matrix remodeling factors, such as Timp1 (Figure 5C, 5F). There was no change in expression upon Chol-MCT1-siRNA treatment, while a significant increase was observed upon GN-MCT1-siRNA treatment. This trend was similar to collagen expression in both cases. Regarding the different types of collagen, instead of measuring each individual type of collagen, we conducted Sirius red and trichrome staining, which enabled us to detect multiple types of collagen simultaneously (Figure 5G, Figure 7D).

b. The authors use multiple animal models including cell specific KO to conclude that stellate cell MCT-1 inhibition decreases COL-1. However, the mechanisms behind this reduced expression of COL-1 are not discussed or explored, making it descriptive.

RESPONSE: We agree that the mechanisms involved are not fully defined but have added new data (Figure 8, Supplement Figure 6) and text to discuss possibilities.

c. Different types of diets are used in this study which could impact lactate availability. Choline deficiency diets are reported to cause weight loss, and importantly have none of the metabolic features of human NASH. Therefore, their utility is doubtful, especially for this study which proposes to investigate if metabolic dysregulation and substrate availability could be a tool for therapy.

RESPONSE: Unfortunately, none of the rodent models used to study NASH completely replicate the condition in human patients, each having its own set of advantages and drawbacks. In line with the concern raised by reviewer #2, there has been a shift away from the use of severely detrimental methionine and choline-deficient diets in contemporary NASH research. Instead, diets that combine methionine and other amino acids with cholinedeficient diets, in conjunction with high-fat diets, have become more popular. The diet we employed in our study consists of high-fat diet combined with choline-deficient diets. We believe that our findings, which are consistent and established across two distinct NASH pathogenesis models and genetic backgrounds, lend additional robustness to our results.

d. Hepatocyte specific MCT-1 KO mice seem to have increased COL-1 production, despite no noticeable difference in hepatocyte steatosis. The reasons for this are not discussed. Fibrosis in NASH is thought to be from stellate cell activation secondary to signals from hepatocellular damage. There is no evidence that there was a difference in either of these parameters in the mouse models used.

RESPONSE: While lipotoxicity-driven liver damage remains a central aspect of NASH pathogenesis, the traditional two-hit theory has become less tractable, giving way to the multi-hit theory in the NASH field. The current debate revolves around whether steatosis is a decisive factor and requirement for NASH fibrogenesis. Our previous publication (Yenilmez et al., 2022, Mol Ther) demonstrated that nearly complete resolution of steatosis did not prevent other NASH features like inflammation and fibrosis, indicating the existence of multiple factors beyond steatosis in NASH pathogenesis. We believe that steatosis and fibrosis influence each other but can also develop independently.

e. The authors report that serum lactate levels did not rise after MCT-1 silencing, but the reasons behind this are unclear. There is insufficient data about lactate production and utilization in this model, which would be useful to interpret data regarding steatosis and fibrosis development. For example, does the MCT-1 KO prevent hepatocyte and stellate cell net import or export of lactate? What is the downstream metabolic consequence in terms of pyruvate, acetylCoA and the NAD/NADH levels. Does the KO have downstream effects on mitochondrial TCA cycling?

RESPONSE: Due to both biological and technical challenges (which are described in the new draft), conducting a comprehensive metabolomics study comparing hepatocyte MCT1 KO to hepatic stellate cell MCT1 KO was not feasible. It is important to note that MCT1 can also transport other substrates that are often overlooked, including pyruvate, short-chain fatty acids, and ketone bodies. Also, in addition to MCT1, there are at least two other functional isoforms of MCT: MCT2 and MCT4. Regrettably, due to these biological and technical complications, conducting a comprehensive metabolomic analysis is extremely complicated and difficult to interpret. Nevertheless, some insights are gained from a study involving MCT1 chaperone protein Basigin/CD147 knockout (KO) mice in a high-fat diet- induced hepatic steatosis model. Basigin acts as an auxiliary protein for MCT1, and its absence leads to improper localization and stabilization of MCT1, effectively simulating a state of MCT1 deficiency. In this context, hepatic lactate levels were reduced by half, and other metabolites such as pyruvate, citrate, α-ketoglutarate, fumarate, and malate were significantly decreased. While we must exercise caution when extrapolating these findings to our MCT1 study, they suggest that multiple metabolites, particularly pyruvate, may play a crucial role in the context of MCT1 deficiency.

f. MCT-1 protein expression is measured only in the in vitro assay. Similar quantitation through western blot is not shown in the animal models.

RESPONSE: We monitored MCT1 protein expression with either Western blot (Fig 2D, 2E (in vitro)) or immune-histology (Fig 4B, 4C (in vivo, ob/ob + GAN diet NASH model), Sup Fig 5F, 5G (in vivo, MCT1 f/f + CDHFD model)).

**Reviewer #3 (Public Review):**
A major finding of this work is that loss of monocarboxylate transporter 1 (MCT1), specifically in stellate cells, can decrease fibrosis in the liver. However, the underlying mechanism whereby MCT1 influences stellate cells is not addressed. It is unclear if upstream/downstream metabolic flux within different cell types leads to fibrotic outcomes. Ultimately, the paper opens more questions than it answers: why does decreasing MCT1 expression in hepatocytes exacerbate disease, while silencing MCT1 in fibroblasts seems to alleviate collagen deposition? Mechanistic studies in isolated hepatocytes and stellate cells could enhance the work further to show the disparate pathways that mediate these opposing effects. The work highlights the complexity of cellular behavior and metabolism within a disease environment but does little to mechanistically explain it.

RESPONSE: Described above to Reviewer #2

The observations presented are compelling and rigorous, but their impact is limited by the nearly complete lack of mechanistic insight presented in the manuscript. As also mentioned elsewhere, it is important to know whether lactate import or export (or the transport of another molecule-like ketone bodies, for example) is the decisive role of MCT1 for this phenotype. Beyond that, it would be interesting, albeit more difficult, to determine how that metabolic change leads to these fibrotic effects.

RESPONSE: Described above to Reviewer #2

Kuppfer cells are initially analyzed and targeted. These cells may play a major role in fibrotic response. It will be interesting to determine the effects of lactate metabolism in other cells within the microenvironment, like Kuppfer cells, to gain a complete understanding of how metabolism is altered during fibrotic change.

RESPONSE: To address the potential involvement of inflammatory cells, we added new data to the manuscript (Supplement Figure 4). Given the distinct hepatic cellular distribution of Chol-MCT1-siRNA and GN-MCT1-siRNA, the opposite fibrogenic phenotype observed may be attributed to MCT1’s role in non-hepatocyte cell types such as the inflammatory Kupffer cells and the fibrogenic hepatic stellate cells. To determine which hepatic cell type drives the opposite fibrotic phenotypes, we first hypothesized that GN-MCT1-siRNA activates M2 pro-fibrogenic macrophages more than Chol-MCT1-siRNA does. The representative M1/ M2 macrophage polarization gene markers were monitored in Kupffer cells. However, GN-MCT1-siRNA treatment caused comparable M1/M2 macrophage activation levels to Chol-MCT1-siRNA treatment (Supplement Figure 4A, 4B). These data suggest that the opposite fibrotic phenotypes caused by the different siRNA constructs are not due to M1/M2 macrophage polarization.

The timing of MCT1 depletion raises concern, as this is a largely prophylactic experiment, and it remains unclear if altering MCT1 would aid in the regression of established fibrosis. Given the proposal for translation to clinical practice, this will be an important question to answer.

RESPONSE: Agree these are important experiments for future evaluation.

**Reviewer #1 (Recommendations For The Authors):**
As above, in general, the conclusions match the data presented. The one exception is the authors discussion point that these data show the importance of lactate flux in fibrosis. As MCT1 has other substrates, it does not seem this is definitively due to lactate flux. It would be helpful to have additional experiments to clarify mechanism by which loss of hepatocyte MCT1 leads to increased fibrosis, while loss of HSC MCT1 reverses this finding. This may aid in concluding that altered fibrosis is in fact due to lactate flux in these cell types.

RESPONSE: Described above to Reviewer #2

In addition, it is unclear why the authors switched NASH models for the two tools generated (GAN diet for siRNA, CDHFD for AAV). Similarly, methodology to assess fibrosis switched between these two experiments - i.e. Sirius Red staining for siRNA-treated GAN diet-fed mice vs. Trichrome staining for AAV-transduced CDHFD-fed mice. These changes make it difficult to perform cross-comparisons of the data, to explain (for example), why GN-siRNA to Mct1 reduced body weight but AAV8-TBG-Cre did not. Similarly, GN-siRNA increased liver Col1a1 protein but AAV8-TBG-Cre did not. These differences could be explained by model system, or tool efficacy/off-target effects.

RESPONSE: We agree that different model systems can explain difference in results, but there is also an advantage of using different models and various methodologies as preclinical tests of consistency of data on NASH under different conditions. There are no perfect mouse models for human NASH.

Phenotyping is also incomplete for the latter experiment, in particular amount of liver lipid content –

RESPONSE: We estimated lipid content by H&E (Fig 6E, F). In some experiments, we focused mostly on COL1 protein expression, as this rather than mRNA is the functional aspect of fibrosis.

**Reviewer #2 (Recommendations For The Authors):**
This study could benefit from standardization of the types of diet used across all animal models and a more comprehensive focus on the metabolic/substrate availability and utilization aspects of NAFLD and NASH affected in the mouse models with MCT-1 dependent lactate transport deficiency. Since hepatic fibrogenesis in NASH is impacted by signals following hepatocyte damage, the extent of cell death in these models could also be better characterized.

RESPONSE: Our ALT data provides indirect insight into hepatocyte damage. Our histology images did not reveal significant changes in cell morphology or integrity and there were no notable changes in caspase protein levels.

Other comments:In Fig 4G, there is an increase in the number of lipid droplets with Chol- MCT-1 siRNA compared to GN-MCT1-sirRNA, suggesting that the stellate cell component might be responsible for this finding. The possible reasons for this are not discussed.

RESPONSE: The effects in Fig 4G were exceedingly small and there is no difference in total TG in these experiments, so it is hard to interpret these data and provide logical explanations.

In Fig 5A. A western Blot for aSMA and COL 1 is shown but the sample labeling is unclear i.e, do the lanes belong to different mice of the same condition? HFD mice vs Ctr mice?

RESPONSE: Both groups of ob/ob mice were fed a GAN diet. The graph in Fig 5 is a direct comparison between NTC-siRNA and MCT1-siRNA. To enhance clarity, this is indicated in the figure legends, and the data in Fig 5 is a continuation of the data presented in Fig 4

In Fig 5E, COL1 densitometry data should also be provided for non-silenced mice on HFD and Chow diet for appropriate comparison

RES\PONSE: Both groups of ob/ob mice were fed a GAN diet. The graph in Fig 5represents a comparison between NTC-siRNA and MCT1-siRNA. It's important to note that, typically, ob/ob mice fed either a chow diet or a high-fat diet do not exhibit fibrogenic phenotypes within this time frame (3 weeks of dietary intervention).

There are many mis-statements throughout the text.Page 6 - "MCT1 silencing significantly inhibited Tgf1β-stimulated ACTA2 mRNA expression as well as collagen 1 protein production" but it is not stated that CO1A1 mRNA is unchanged in Fig 1C.

RESPONSE: We observed no change in CO1A1 mRNA levels (Fig 1C), so we focused on collagen 1 protein production (Fig 1B) on page 6. Given the consistent trend observed in Chol-MCT1-siRNA (Fig 5C), we proposed the possibility of MCT1's influence on collagen translation or protein turnover on page 11.

Page 7- ".......our Chol-MCT1-siRNA does not require transfection reagents as it is fully chemically modified". What does fully chemically modified mean and why does this mean in terms of transfection efficiency.

RESPONSE: One of the primary challenges in utilizing RNAi as a therapeutic approach has been the effective in vivo delivery strategy, particularly concerning stability and longevity against systemic nucleases. Recent developments in siRNA duplex chemical modification strategies, such as 2-Fluoro and 2-O-Methyl ribose substitutions, as well as phosphorothioate backbone replacements, have addressed these challenges Please see Figure 3. In our current study, we employed 'chemically fully modified' siRNA, featuring several key modifications: (1) every single ribose is chemically modified to 2-F or 2-OMeribose, (2) phosphorothioate backbone replacement, (3) 5'-end of the antisense strand modification to (E)-Vinyl-phosphonate, and (4) 3'-end of the sense strand linkers such as Cholesterol or Tri-N-Acetyl-galactosamine. These chemical enhancements significantly improve transfection efficiency, longevity, and selectivity, setting it apart from traditional siRNA lacking such chemical modifications. A prior study from the Khvorova lab has demonstrated substantial efficiency differences between partially and fully modified siRNA in vivo.

Page 7- the results present for Fig 2 ignores Fig, 2C, if this is important it needs to be described if not, please delete.

RESPONSE: The dose-response potency results, crucial for identifying the most potent Chol-MCT1-siRNA compound, are depicted in Figure 2C. The wording "(Figure 2C)" has been inserted in the sentence as follows. “The silencing effect on Mct1 mRNA was monitored after 72 hours (Figure 2B). Several compounds elicited a silencing effect greater than 80% compared to the NTC-siRNA. The two most potent Chol-MCT1-siRNA, Chol-MCT1-2060 (IC50: 59.6nM, KD%: 87.2), and Chol-MCT1-3160 (IC50: 32.4nM, KD%: 87.7)(Figure 2C) were evaluated for their inhibitory effect on MCT1 protein levels (Figure 2D, 2E). Based on its IC50 value and silencing potency, Chol-MCT1-3160 construct was chosen for further studies in vivo (Table 2).”

Supplement Fig 1A-F should be analyzed by multiple comparisons not by paired t-tests.

RESPONSE: We performed t-tests for every comparison between two groups. However, for Sup Fig 1A-F, which involved a comparison among three different groups, we applied oneway ANOVA.

The x-axis in supplement Fig 2A and B are not labeled, and I assume are in weeks. The Fig 2B x-axis numbers also mis-labeled and should also be 0-3 and not 10-13.

RESPONSE: The x-axis is now appropriately labeled.

Page 10 - the description of supplement Fig 4A is not accurate. Srebf1 mRNA is unchanged by the GN-MCT1-siRNA treatment and Mlxipl mRNA is unchanged by Chol-MCT1-siRNA treatment. Is this total Mlxipl mRNA or can you distinguish between the alpha and beta variants.

RESPONSE: We adhered to NCBI nomenclature, where 'SREBP1' and 'ChREBP' represent proteins, not mRNA. The Mlxipl mRNA we tested pertains to total Mlxipl mRNA. Original draft shown below.

“To investigate the underlying mechanism by which lipid droplet morphological dynamics change, we monitored the effect of hepatic MCT1 depletion on DNL-related gene expression. Both GN-MCT1-siRNA and Chol-MCT1-siRNA strongly decreased the mRNA and protein levels related to representative DNL genes (Supplement Figure 4A-4D). Intriguingly, both modes of hepatic MCT1 depletion also inhibited expression of the upstream regulatory transcription factors SREBP1 and ChREBP.”

There are no molecular weight markers in supplement Fig 4C and D. Is the Srebp1c blot for the nuclear or precursor form?

RESPONSE: The Srebp1c blot presented represents the precursor form. I have edited the figure legend accordingly. It's worth noting that the cleaved form of Srebp1c either exhibited significantly lower expression compared to its precursor form or displayed comparable expression between the control group and the MCT1 depletion group.

Changes in mRNA and protein do not always reflect changes in activity (allosteric regulation). If you want to draw any conclusions about de novo lipogenesis you need to directly measure fatty acid synthesis rates from a carbohydrate precursor.

RESPONSE: We completely agree. Therefore, in the current study, we emphasized two key points: (1) hepatic MCT1 depletion affects the expression levels of representative DNL genes, and (2) however, this regulation was insufficient to resolve the steatosis phenotypes in our NASH model. We have added the text “while recognizing that the decreased expression of DNL genes does not necessarily indicate inhibited fatty acid synthesis rate” on page 15.

**Reviewer #3 (Recommendations For The Authors):**
Figure 1 - Are there changes to fibroblast phenotype with TGF-beta stimulation and are these changes reversed with MCT1 siRNA-mediated silencing, or is this purely an expression phenomenon?

RESPONSE: This study was designed to assess the preventative effect of MCT1 silencing on Tgf1β-induced fibrosis, rather than a reversal study. As detailed in the methods section, LX2 cells were initially cultured in DMEM/high glucose media with 2% FBS. The following day, we transfected the cells with either NTC-siRNA or MCT1-siRNA (IDT, cat 308915476) using Lipofectamine RNAi Max (ThermoFisher, cat 13778075) for 6 hours in serum-reduced Opti-MEM media (ThermoFisher, cat 31985062). Subsequently, the cells were maintained in serum-starved media, with or without 10ng/ml of recombinant human Tgf1β (R&D Systems, cat 240-B/CF), for 48 hours before harvesting.

Is lactate import/export itself responsible for this phenotype? It is presumed that MCT1 depletion alters import/export of lactate and subsequently modulates this phenotype, but this is never shown experimentally. Does lactate accumulate in these cells or in the medium in culture? The foundation of the paper rests on this hypothesis, so we believe that this is critical to establish. This is particularly relevant as MCT1 has been proposed to function primarily as a lactate importer, so the availability of medium lactate could be easily modulated to determine whether that mimics MCT1 loss.

RESPONSE: To address the underlying mechanism of MCT1/Lactate in stellate cells, we added a new figure to the manuscript (Figure 8). We had previously conducted an experiment to determine whether MCT1 depletion in LX2 cells in vitro influences extracellular lactate concentrations in DMEM/high glucose (25mM glucose) media supplemented with 1mM sodium pyruvate but without sodium lactate. Interestingly, we found no significant difference in extracellular glucose and lactate concentrations, which remained at 25mM and 5mM, respectively. These concentrations were comparable between groups, regardless of MCT1 loss. Additionally, we investigated the effects of MCT1 silencing in the presence of potent fibrogenic inducer TGF-β1. Intriguingly, MCT1 depletion effectively prevented TGF-β1-induced collagen production, irrespective of lactate (+/- pyruvate) supply in the media. LX2 cells with MCT1 depletion exhibited reduced collagen 1 production when lactate was solely generated by endogenous glycolysis (Figure 8F) and when exogenous lactate was supplied (Figure 8G).

Figure 2 - It is compelling that the Chol-MCT1-siRNA compounds are effective at targeting MCT1. However, is it clear how specific the siRNA target is? Are other MCT genes affected as well (if the siRNAs target areas of homology, for example)? Given that this siRNA strategy is used going forward and proposed as a therapeutic, it would be important to discuss and perhaps characterize off-target effects. A simple BLAST search for homology for the chosen siRNAs could help answer this question.

RESPONSE:

1. We designed the siRNA to specifically avoid any potential off-target effects on MCT1's 14 isoforms, and this approach aligns with the results obtained from the NCBI-BLAST analysis.

2. While there are 14 isoforms of MCTs, only the first four are functional. To assess the off-target effect of Chol-MCT1-siRNA on MCT2 and MCT4 (MCT3 was excluded due to its limited expression in retinal pigment epithelium), we conducted in vivo experiments in ob/ob mice, which demonstrated a highly selective MCT1 silencing effect. We have also included MCT1, MCT2, and MCT4 rt-qPCR data in the manuscript (Supplement Figure 2A, 2B).

3. We plan to further optimize and validate the human MCT1-targeting siRNA sequence for use in humanized mouse studies. It's important to note that the MCT1-siRNA used in this study was designed for mice.

Supplemental Figure 1 - brain would be one other highly metabolic tissue wherein it would be important to show lack of activity/accumulation.

RESPONSE: Undoubtedly, the brain is one of the most metabolically active tissues, playing a pivotal role in regulating signaling pathways and metabolism in other tissues. However, it poses a significant challenge in terms of targeting due to the presence of the blood-brain barrier (BBB). Overcoming BBB penetration remains one of the foremost challenges in the field of therapeutic siRNA delivery. For many therapeutic oligonucleotides, including Cholesterol-conjugated siRNAs, systemic administration alone is normally insufficient to achieve BBB penetration. Direct local injection or transient disruption of the BBB is normally required.

Figure 4 - The image shown for chol-MCT1-siRNA seems to show variation in lipid droplet size. Is this just this single image? The authors quantify smaller lipid droplets in this group, so the image may not be representative as there are many large droplets. Ultimately, additional mechanisms as to how alterations in lactate metabolism could mediate this phenotype are missing. This hypothesis also rests upon the assumption that MCT1 is modulating lactate, which is not shown experimentally, as discussed above.

RESPONSE: We changed the representative images (Fig 4B). We agree this aspect of the study is not resolved, and we have related text in the manuscript on this point: “neither GNMCT1-siRNA nor Chol-MCT1-siRNA decreased total hepatic TG levels (Figure 4H), although quantitative analysis of H&E images showed a small decrease in mean lipid droplet size and increased number of lipid droplets upon MCT1 silencing (Figure 4F, 4G). These data suggest the possibility that hepatic MCT1 depletion either (1) inhibits formation or fusion of lipid droplets, or (2) enhances lipolysis to diminish lipid droplet size.”

Figure 5 provides evidence that Chol-MCT1-siRNA expression decreases fibrosis but this is attributed to the effects on stellate cells. While GN-MCT1-siRNA and subsequent MCT1 silencing in hepatocytes has an opposite effect. The cell population that is not discussed, however, is the Kupffer cell. Could MCT1 silencing in this cell population be mediating part of the phenotype observed? How does MCT1 silencing affect Kupffer cell phenotype and activity?

This extends into Figure 6 where Kupffer cells are not given consideration in targeted experiments.

RESPONSE: Described above to Reviewer #3

Figure 6 and 7 use a different model to show that stellate cell depletion of MCT1, specifically, decreases collagen 1 protein levels in NASH, which reinforces the authors claims. Given the cell specificity of this experiment, it is more compelling data. It would be nice to show that Kupffer cell depletion of MCT1 does not have any affect or perhaps show that it does.

RESPONSE: We agree, but Kupffer selective depletion is not possible to do with this siRNA technology. Please see the response above as our most recent attempt to address this question.

Figure 7 shows that even with decreased collagen deposition, there is no effect on liver stiffness or chronic liver injury as measure by ALT. This may suggest that the decreased level of fibrosis is either not significant to overall clinical outcome or that there are other fibroinflammatory mechanisms compensating for lack of COL1 deposition. Is there increased reticulin fibrosis when MCT1 is knocked down? This could be assessed with IHC or monitoring type 3 collogen (COL3A1).

RESPONSE: Reticulin fibrosis results from the excessive deposition of reticular fibers, primarily composed of type 3 collagen. However, based on our observation of trichrome staining in whole liver histology data (Fig 7D-E), which exhibited nearly identical trends to collagen type 1 expression (Fig 7A-C), it seems unlikely that type 3 collagen compensated for the decrease in type 1 collagen protein expression upon hepatic stellate cell MCT1 KO. We plan to perform detailed analysis of a more comprehensive list of ECM proteins including type 3 collagen in our humanized mouse model with engrafted human liver cells in future experiments.

Additional considerations:It may be useful to know if inhibition of fibrosis affects survival/progression in these NASH models over a longer timeframe, although this may understandably be beyond the scope of the current work.The timing of MCT1 depletion is prophylactic and given the proposal to translate this research, it would be important to determine whether MCT1 inhibition reversed fibrosis, and if so, by what metabolic mechanism?

RESPONSE: We have observed that extending the duration of the NASH model increases the likelihood of hepatocarcinoma development. Exploring the aim to include survival and disease progression as well as reversal of fibrosis would be important in future experiments.

Summary of new Figures and Figures modified:

Fig 1B: added "and" (significance) between the first and the third group, and the second and the last group.Fig 4B: replaced images with more representative ones as the mean lipid size was questioned by the reviewer.Fig 7D: made the images bigger (original images cropped and enlarged → 5X)Fig 8: newly created to explain the underlying pathway of lactate, and MCT1 regulating collagen production. Please find the results sections.Sup fig 2A, B: newly added to show our compounds’ selective silencing effect. - Sup Fig 2C-D: Added missing x-axis (moved from previous Figure 2A, 2B) - Sup Fig 2E-F: moved from sup Fig 3 not to have too many sup figures.Sup Fig 3C-D: showed both precursor and cleaved form of SREBP1 bands as requested (moved from previous sup Figure 4)Sup Fig 4: newly created, as questioned many times for the effect on Kupffer cells or other inflammatory cells.Sup Fig 6: newly created to explain the potential underlying mechanism of MCT1 depletion on collagen production.Sup Fig 7: moved from previous sup Fig 6.Sup Fig 8: moved from previous sup Fig 7.